# Large Igneous Province thermogenic greenhouse gas flux could have initiated Paleocene-Eocene Thermal Maximum climate change

Stephen M. Jones [1]*, Murray Hoggett [1], Sarah E. Greene [1] & Tom Dunkley Jones [1]

Large Igneous Provinces (LIPs) are associated with the largest climate perturbations in Earth's history. The North Atlantic Igneous Province (NAIP) and Paleocene-Eocene Thermal Maximum (PETM) constitute an exemplar of this association. As yet we have no means to reconstruct the pacing of LIP greenhouse gas emissions for comparison with climate records at millennial resolution. Here, we calculate carbon-based greenhouse gas fluxes associated with the NAIP at sub-millennial resolution by linking measurements of the mantle convection process that generated NAIP magma with observations of the individual geological structures that controlled gas emissions in a Monte Carlo framework. These simulations predict peak emissions flux of 0.2–0.5 PgC yr$^{-1}$ and show that the NAIP could have initiated PETM climate change. This is the first predictive model of carbon emissions flux from any proposed PETM carbon source that is directly constrained by observations of the geological structures that controlled the emissions.

[1] School of Geography, Earth & Environmental Sciences, University of Birmingham, Edgbaston, Birmingham B15 2TT, UK. *email: s.jones.4@bham.ac.uk

Temporal associations between Large Igneous Provinces (LIPs) and perturbations to global climate, ecosystems and the carbon cycle occur throughout Mesozoic time, from the Permo-Triassic mass extinction (the most devastating in Earth's history) through multiple Ocean Anoxic Events[1,2]. They imply that greenhouse gases released directly by LIPs can initiate global change that persists over $10^4$–$10^5$ years. The Paleocene–Eocene Thermal Maximum (PETM) is the largest natural climate change event of Cenozoic time, and an important yardstick for anthropogenic climate change[3,4]. During PETM initiation, release of 0.3–1.1 PgC $yr^{-1}$ of carbon as greenhouse gases to the ocean–atmosphere system[4–6] drove 4–5 °C of global warming[7] over a short period (<20,000 years)[5,8–10]. Although the North Atlantic Igneous Province (NAIP) LIP and the PETM are closely coincident in time[11–13], the rate and duration of NAIP carbon emissions have not yet been reconciled with the <20 kyr onset of PETM climate change[8,9,14,15].

Several mechanisms by which the NAIP might have supplied carbon-based greenhouse gases to the ocean–atmosphere system have been proposed. Carbon dioxide was released from NAIP magma[16]. Methane and carbon dioxide were generated by thermal maturation of organic material within sedimentary rock next to shallow igneous intrusions[17]. Regional seabed uplift may have released methane, depending on the size of the pre-PETM subseafloor methane hydrate inventory[18]. This seafloor uplift would have altered oceanic circulation[19] and regional climate[20], which could have led more indirectly to pulsed greenhouse gas release[21]. However, all these theories struggle because none has been shown to deliver a peak in gas emissions flux whose duration matches the <20 kyr timeframe of PETM onset[8,9,14,15]. Consequently, the NAIP has typically been relegated to a driver of longer term ($10^5$–$10^6$-year) background warming, which perhaps triggered more rapid release of carbon from other temperature-sensitive near-surface reservoirs around the globe[22–24].

Thermogenic methane produced by shallow igneous sills (subhorizontal sheets of magma) and released to the atmosphere or shallow ocean through hydrothermal vents is the most likely source of the large mass of carbon (up to 13,000 PgC)[5,13] required to explain the entire PETM[17]. Environmental change records imply that carbon emissions rates reached 0.3–1.1 PgC $yr^{-1}$ for several thousand years across the PETM onset[4–6]. Carbon emissions rates generated by representative individual NAIP sill–vent systems have not been reported. However, a single sill can generate total emissions of 0.01–2 PgC, based on existing thermal aureole modelling[17,25] applied to individual NAIP sill surface area measurements[26] of 5–50 $km^2$. Since the characteristic cooling time period for these sills is 100–1000 years[27], the mean emissions flux from an individual sill is unlikely to exceed 0.002 PgC $yr^{-1}$. Many sill–vent systems must therefore have supplied greenhouse gas simultaneously if NAIP sill province emissions are to explain the PETM.

A crucial parameter to bridge this gap is $\tau_{repeat}$, the typical time period between intrusion of successive sills. $\tau_{repeat}$ must be significantly less than the typical cooling time period for one sill (i.e. <100 years) for many sill–vent systems to be active simultaneously. The simplest way to estimate $\tau_{repeat}$ is to divide the sill province duration by the total number of sills. There are 11,000–18,000 sill–vent systems within the NAIP[17] but the sill province duration is uncertain. The generally quoted duration for the starting phases of the NAIP and other LIPs is 1–3 Myr[28,29], giving $\tau_{repeat}$ of 56–273 yr. However, 95% of sills intruded simultaneously (at seismic imaging resolution) in individual basins near the NAIP centre, perhaps suggesting 60 kyr for the local province duration[17], which corresponds to $\tau_{repeat}$ of 3 yr. Measuring $\tau_{repeat}$ for the NAIP sill province directly using traditional radiometric or biostratigraphic dating methods would be

a huge undertaking. A fundamental difficulty is that these methods have $10^5$–$10^6$-year resolution, whereas the PETM onset was $10^4$ years or less. In principle, this difficulty might be overcome by statistical modelling of large numbers (>100) of radiometric ages of sills across entire NAIP footprint, as has been shown for the Karoo LIP[30]. Unfortunately, much of the NAIP lies within the deep-water oil exploration frontier; it could be many decades before sufficient sill samples from boreholes become available.

Here we use our concept of $\tau_{repeat}$ to demonstrate for the first time that the NAIP sill province could have intruded sufficiently rapidly to initiate the PETM. We tackle the problem in two stages. First, we determine the $\tau_{repeat}$ that would be required for the NAIP sill province to match carbon emissions rates that have been independently shown to initiate the PETM. Secondly, we demonstrate that such $\tau_{repeat}$ values were likely achieved during the most intense phase of NAIP sill intrusion. These two steps required development of several new databases and calculation procedures. We began by developing a new parameterisation of thermogenic and magmatic carbon emissions from individual sill–vent systems of known dimensions intruding a host of known organic content. We then assembled a large new database of NAIP sill and host-rock observations. We also developed a method to determine the carbon emissions from an entire sill province by summing emissions from many sill–vent systems. Together, these components allow Monte Carlo simulations of geologically plausible combined carbon emissions from a sill province when the $\tau_{repeat}$ is specified a priori, which show that $\tau_{repeat}$ of 2–6 yr would be required to initiate the PETM. To complete the second stage in our argument, we developed a novel alternative to dating volcanic products that considers instead the mantle convection process that generated the sill province magma: we derived an expression linking $\tau_{repeat}$ to mantle plume flux (Fig. 1). We then assembled new databases of mantle plume flux measurements and the geographical distribution of sills across the NAIP. We use this information to estimate how $\tau_{repeat}$ varied throughout the emplacement history of the NAIP sill province, and show that $\tau_{repeat}$ could have dropped below 5 yr and initiated the PETM. Thus, we present the first predictive, mechanistic model of carbon emissions flux from a LIP.

## Results

**Thermogenic carbon emissions flux parameterisation procedure.** Although there is a large body of work on contact metamorphism next to igneous sheets[25], no studies provide time-series of methane emissions flux that we can use for stochastic modelling and few studies model methane generation directly. We require a rapid method to calculate carbon flux from individual sills with a wide range of dimensions because the calculation must be repeated over 10,000 times to simulate combined emissions from the entire NAIP, and hundreds of simulations of the entire province are needed to determine uncertainty bounds. We therefore developed a new parameterisation of thermogenic reaction kinetic modelling results. We first carried out a series of coupled thermal and reaction kinetic calculations that spanned the observed ranges in sill dimensions and emplacement depth (Fig. 2). These calculations were done in one spatial dimension (distance perpendicular to the sill) to give the necessary flexibility to scale the results to the observed ranges of sill and host-rock dimensions during stochastic modelling.

We only considered thermogenic reactions that produce methane from organic matter. Almost all of the carbon produced from the NAIP sill province host rocks (mainly mudrock) is generated via these reactions[25]. We calculated time-series of methane gas generation directly, rather than tracking vitrinite

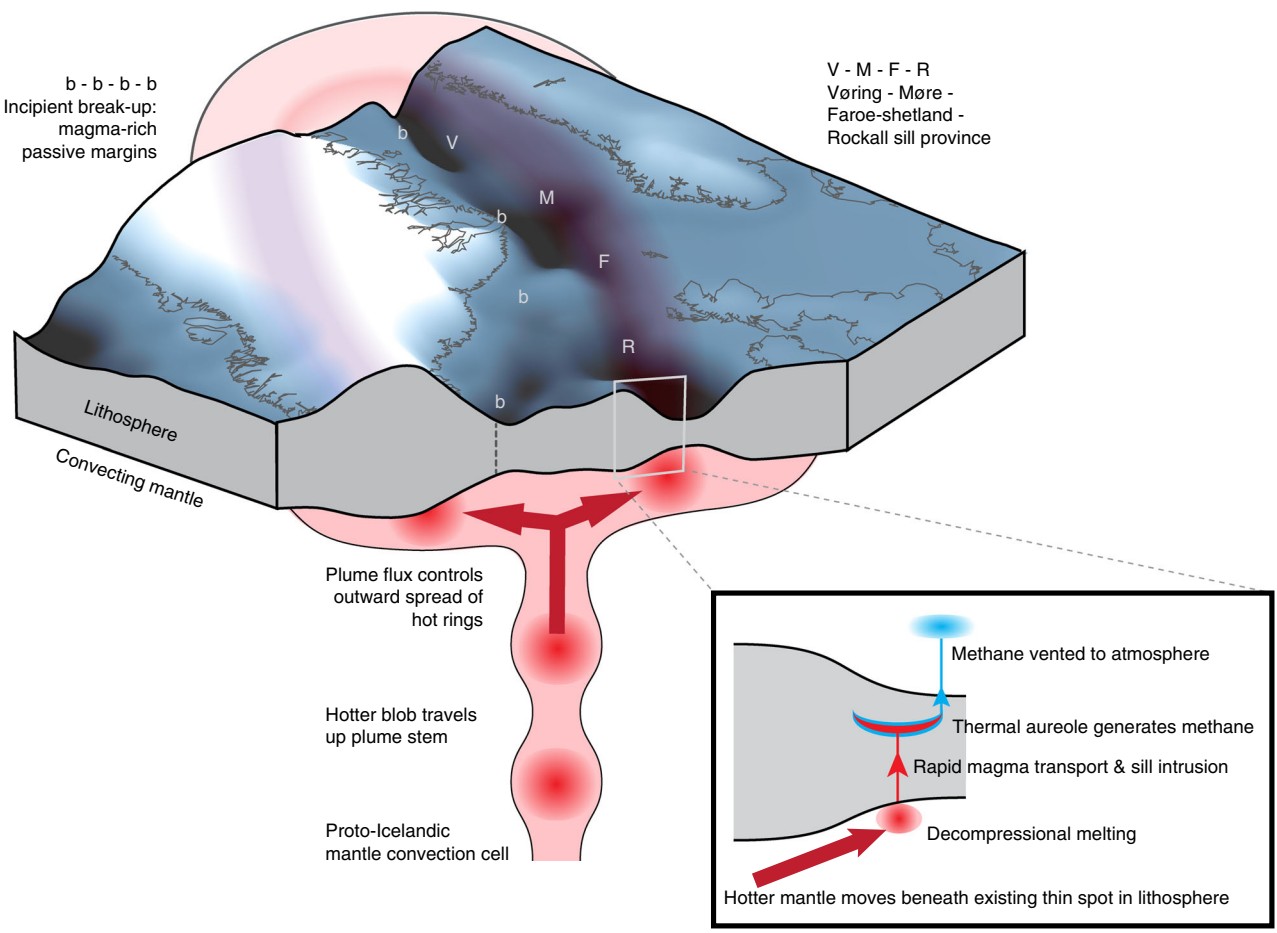

**Fig. 1** New framework for determining greenhouse gas emissions from the North Atlantic Igneous Province. Initiation of the Icelandic mantle convection cell occurred during early to middle Paleocene time, accompanied by volumetrically minor igneous activity. By late Paleocene time, a mantle plume head comprising anomalously hot mantle lay beneath the lithosphere of NW Europe and Greenland, fed by a conduit beneath eastern Greenland. Flow within all upper mantle convection cells is unsteady, or pulsing. Pulses are patches of mantle with particularly high temperature anomalies that travel up the plume conduit and spread outwards within the plume head as rings. The hottest of these pulses travelled up the plume conduit in latest Paleocene time and spread through the plume head in earliest Eocene time. When this thermal pulse travelled beneath thin lithosphere, the reduction in depth led to pressure-release partial melting. The rate of magma generation depended on the mantle plume flux. Most thin-spot melting occurred beneath the chain of deep-water sedimentary basins comprising the Rockall, Faroe-Shetland, Møre and Vøring Basins (labelled R, F, M and V, respectively), which had formed by rifting during mid Cretaceous time. Basaltic magma generated at the base of the lithosphere separated rapidly from the mantle source and travelled rapidly upwards. Much of this magma stalled within the crust as roughly horizontal, sheet-like intrusions called sills, known collectively as the NAIP sill province. The sills heated the surrounding sedimentary rock and converted trace amounts of organic carbon to thermogenic methane. Heat from the sills also boiled the pore fluid within the host rock, which fractured the rock and created hydrothermal vents. Pore-water convection carried the methane to the ocean and atmosphere. Additional carbon emissions comprising predominantly magmatic carbon dioxide are associated with break-up between Europe and Greenland to form the NAIP magma-rich passive margins (labelled b-b-b-b). These emissions likely have lower flux in comparison with peak thermogenic methane emissions from the NAIP sill province. Thus, mantle plume flux is the primary control on both magma generation rate and also thermogenic methane emissions rate associated with the NAIP sill province.

reflectance as a proxy for gas generation as in previous studies of the PETM-NAIP sills link[25]. We used a standard oil-industry kinetic model for gas generation that has been calibrated against both laboratory and geological datasets, which bracket the heating rates generated by shallow igneous sills[31] (Supplementary Table 1). Methane generation involves three reaction pathways: refractory (gas-prone) kerogen matures to produce methane; labile (oil-prone) kerogen matures to produce mostly oil and some methane; and oil cracks to produce methane. Vitrinite reflectance reactions can be tracked using the same scheme[32], and all our model results match a compilation of vitrinite reaction aureole data[25].

The results of coupled thermal and reaction kinetic modelling were used in two ways. First, we established that both the carbon emissions flux as a function of time, $q_{therm}(t)$, and also the cumulative mass of carbon generated by thermal maturation adjacent to an igneous sill, $m_{therm}(t)$, can be conveniently described as tapered power laws defined by three parameters: the final mass of generated carbon $M_{therm}$, the emissions decay time $\tau_{decay}$ and a power exponent $p$ (Fig. 3; Methods). $q_{therm}(t)$ decreases and $m_{therm}(t)$ increases as a power law when $t < \tau_{decay}$. $q_{therm}(t)$ tapers to zero and $m_{therm}(t)$ tapers asymptotically to $M_{therm}$ when $t > \tau_{decay}$.

The second use of the new calculations was to generate look-up tables to estimate the parameter set ($M_{therm}$, $\tau_{decay}$, $p$) required to calculate carbon emissions over time from a set of sill dimensions and host-rock characteristics ($A$, $K$, $S$, $Z$, $w_{CH4}$, $\beta$). The sill dimensions are: $A$, surface area; $S$, maximum thickness; $\beta$, a dimensionless parameter to describe how thickness varies; and $Z$, intrusion depth. The host-rock characteristics are: $w_{CH4}$, the

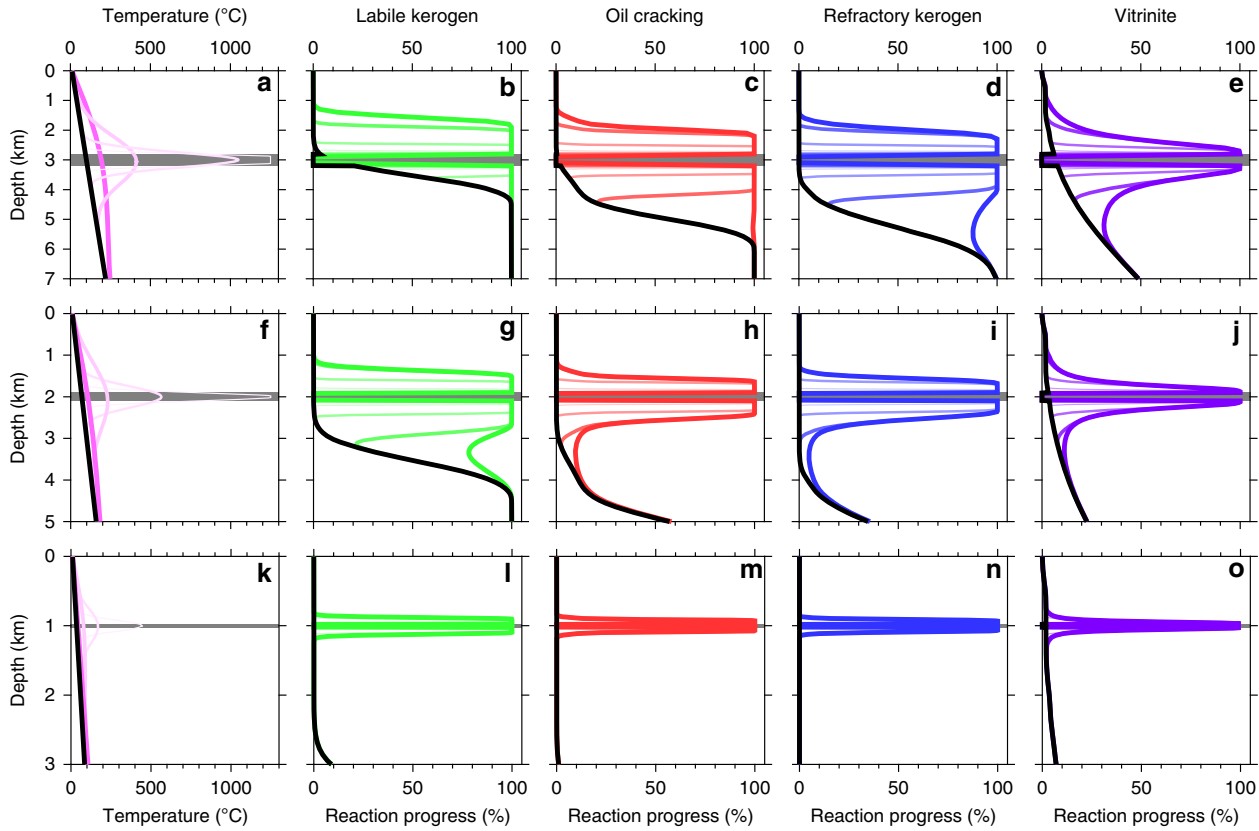

**Fig. 2** Modelling hydrocarbon-generating reactions within thermal aureoles of igneous sills. **a–e** 400-m-thick sill emplaced at 3 km; **f–j** 200-m-thick sill emplaced at 2 km; **k–n** 50-m-thick sill emplaced at 1 km. **a**, **f**, **k** Temperature profiles immediately before sill intrusion (thick black line) and at times after sill intrusion of 100 years (thinnest line, lightest colour), 1000 years, 10,000 years and 100,000 years (thickest line, strongest colour). All other plots show the same time snapshots using corresponding line thicknesses and colour strengths. **b**, **d**, **l** Reaction progress for labile kerogen maturing to oil. **c**, **h**, **m** Reaction progress for oil cracking to methane. **d**, **i**, **n** Reaction progress for refractory kerogen maturing to methane. **e**, **j**, **o** Reaction progress for vitrinite reflectance. The full suite of calculations was done for sill thicknesses of 50, 100, 200, 300 and 400 m at emplacement depths of 0.5, 1.0, 1.5, 2.0, 2.5, 3.0 and 3.5 km.

concentration of organic matter that converts to methane; and $K$, the type of organic matter. Results are stated in terms of the total contact aureole thickness scaled by the sill thickness, $Y$, which is directly proportional to $M_{therm}$, and the emissions decay time scaled by the thermal conductive cooling time, $\tau^\star$ (Supplementary Fig. 1). These results cover the full range of NAIP sill observations for the first time. Sills thinner than 150 m emplaced less than 1.5 km deep have scaled aureoles of $Y = 200–400\%$, in agreement with previous work[25], and emissions decay times similar to their thermal cooling times. As sill thickness and emplacement depth increase to 250 m and 3 km, respectively, $Y$ and $\tau^\star$ increase to 800% and 400%, respectively, because longer solidification times and higher background temperature mean that greater temperatures are maintained for longer near the sill. $Y$ and $\tau^\star$ then decrease with further increases in $S$ and $Z$ because deeper host rocks have already generated and expelled methane by burial maturation prior to sill intrusion. The $p$ exponent values account for the dependence of thermogenic reaction rates on temperature as well as time, so they exceed the $p = 0.5$ indicated by analytical solutions of conductive sill cooling[27].

**Magmatic carbon emissions parameterisation procedure.** Magmatic carbon emissions are easier to model than thermogenic emissions because it is not necessary to track reaction kinetics. Carbon is strongly incompatible in all primary minerals formed during solidification of basaltic sills. Almost all mantle-derived carbon dioxide dissolved in the magma therefore exsolves shortly

before solidification, mixes with the super-critical host-rock pore fluid, and escapes through hydrothermal vents along with thermogenic methane. Our parameterisation is based on the same set of sill and host-rock characteristics used for the thermogenic parameterisation except that $w_{CO2}$, the concentration of carbon dioxide initially dissolved in the magma, is substituted for $w_{CH4}$.

**Sill and host-rock observations database.** A large literature describes igneous sills from the NAIP and around the world but little sill surface area, thickness and emplacement depth information is tabulated. To obtain statistically representative distributions required for stochastic emissions modelling, we measured many hundreds of NAIP sills on two-dimensional (2D) and three-dimensional (3D) seismic reflection data (Fig. 4; Supplementary Fig. 2).

Sill surface areas come from seismic surveys close to the centre of the NAIP. These were checked with reference to 2D data from across the NAIP (e.g. Supplementary Fig. 2) and satellite images of the Karoo sill province. Over 10% of observed sills have surface areas of 100–1000 km$^2$, with potential to generate significant (0.01–0.1 PgC yr$^{-1}$) short-term (c. 1 year) peaks in emissions flux. For both 3D and 2D data, we excluded surface areas < 10 km$^2$ from the histograms used for stochastic modelling because hydrothermal vents are rarely observed associated with such small sills, so their methane likely migrates over a timeframe longer than PETM initiation. Depths of intrusion were obtained from the subset of sills for which the coeval seabed can be

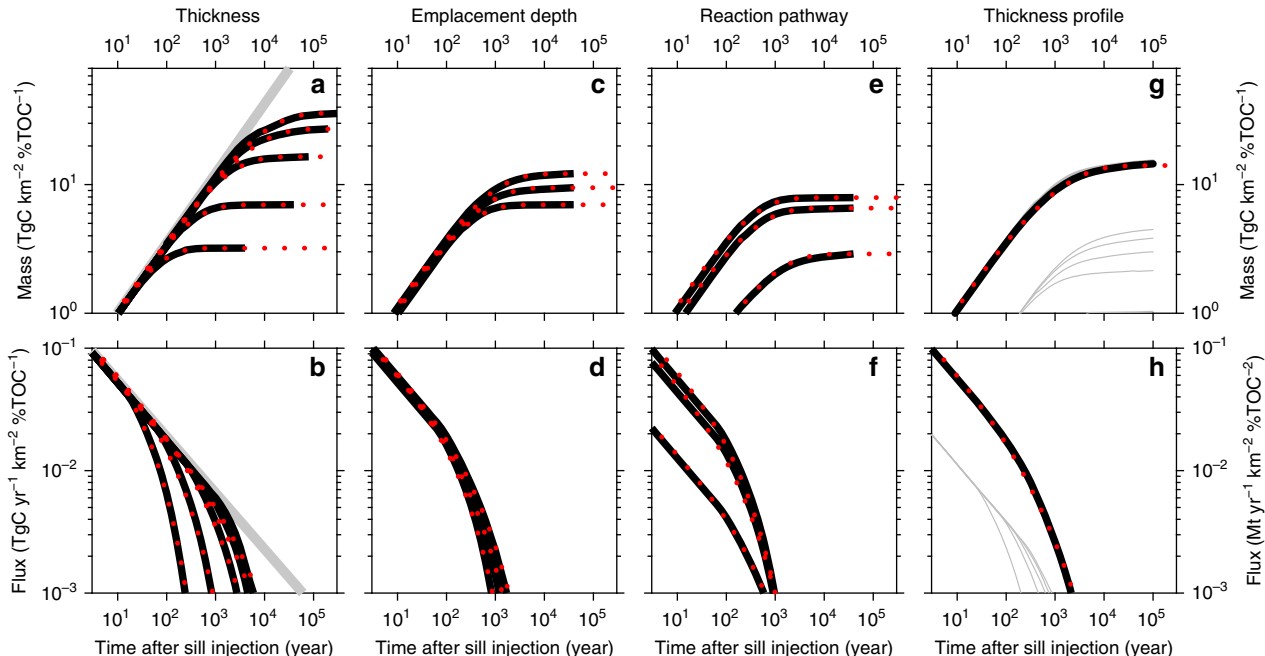

**Fig. 3** Comparison between full carbon emissions modelling results and new carbon emissions parameterisation. Thick black lines represent full thermal and reaction kinetic modelling results; red dotted lines represent new parameterisation. **a**, **b** Mass and flux of carbon emissions from breakdown of labile kerogen for sills of thickness 50, 100, 200, 300 and 400 m, all intruded at 1 km depth. Thick grey line shows a power law with exponent $p = 0.54$ for reference; all calculations in this figure assume a constant host-rock density of 2100 kg m$^{-3}$, appropriate for the mean emplacement depth (1.2 km), in order to illustrate the consistent tapered power-law behaviour of emissions. **c**, **d** Mass and flux of emissions from breakdown of labile kerogen for sills intruded 1, 2 and 3 km deep, all 100 m thick. **e**, **f** Mass and flux of emissions for a 100-m-thick sill intruded 2 km deep from three reaction pathways: breakdown of refractory kerogen; breakdown of labile kerogen to oil and then cracking of oil to gas; and direct from labile kerogen. **g**, **h** Modelling of emissions for a sill of variable thickness profile. A radially symmetrical sill with maximum thickness 200 m and intruded 2 km deep is divided into five annuli of equal surface area. The thickness of each annulus is prescribed by the radial thickness profile parameterisation in Fig. 2. Emissions from these five annuli (thin grey lines) are summed to estimate the total (thick black lines).

identified from associated vents and/or onlap of strata onto forced folds[26]. Measured emplacement depths were corrected for post-emplacement compaction. The observations are well approximated by a normal distribution with mean depth 1.15 km and standard deviation 0.85 km, truncated at the shallow end.

Maximum sill thickness was measured for the subset of sills that show seismic reflections from both top and base. We observe a positive linear correlation ($R^2 = 0.31$) between maximum diameter and maximum thickness. The maximum sill thickness used in stochastic modelling was obtained by first applying the linear regression to the stochastically chosen diameter (assuming a circular sill) and perturbing this value by a thickness selected from a uniform distribution between ±100 m. We measured sill thickness as a function of radius both directly, where top and bottom reflections were visible, and indirectly from forced folds in the contemporary seabed. We find that radial sill profiles of $s/S = (1 - r^2/R^2)^\beta$ with $0.64 < \beta < 1.01$ provide a good match to both sets of observations, where $s/S$ is the local sill thickness scaled by the maximum thickness and $r/R$ is the local radius scaled by the maximum radius.

Host-rock carbon content has already been measured for hundreds of samples of Paleocene and Cretaceous host rocks offshore Norway[33], though no comparable data are yet available for UK and Irish parts of the NAIP sill province. The models presented here assume that organic matter is predominantly labile (oil-producing). This choice of organic matter type does not significantly affect the emissions flux results, but it does slightly improve the fit to carbon isotopic records compared with model host rocks with greater proportions of refractory kerogen.

**Combined LIP sill province carbon emissions**. We developed a method to determine the carbon emissions flux and cumulative emitted carbon from the entire sill province by summing thermogenic and magma degassing emissions from many sill–vent systems when $\tau_{repeat}$ is known. In each NAIP simulation, emissions parameters for all 11,000–18,000 sill–vent systems are derived from observations (Fig. 4) using a Monte Carlo approach. In any one NAIP simulation, initial (maximum) emissions flux from individual sill–vent complexes ranges from 0.0005 to 0.05 PgC yr$^{-1}$ over the year following intrusion (Fig. 5a), compared with the peak flux of 0.3–1.1 PgC yr$^{-1}$ sustained over several millennia required to explain the PETM[4–6]. Emissions decay times for individual complexes range from 5 to 5000 yr (Fig. 5b).

In order to determine the $\tau_{repeat}$ required to explain the PETM onset, we carried out ensembles of NAIP simulations at various constant values of $\tau_{repeat}$. In each ensemble, emissions flux rises over c. 1000 yr to c. 90% of the final value, then rises more slowly to reach a stable mean value by 5–10 kyr. The relationship between long-time mean emissions flux and repeat time is well approximated by

$$Q_{prov} = 1.65/\tau_{repeat}. \tag{1}$$

Thus, we find that a carbon flux capable of initiating the PETM is achieved with $\tau_{repeat}$ between 2 and 6 years (Fig. 4e, f).

**Derivation of link between $\tau_{repeat}$ and mantle plume flux**. We avoided the need for radiometric or biostratigraphic dating of sills by deriving an alternative constraint on $\tau_{repeat}$ based on the mantle convection process that generated the NAIP (Fig. 1). This approach provides confidence because the mantle convection cell

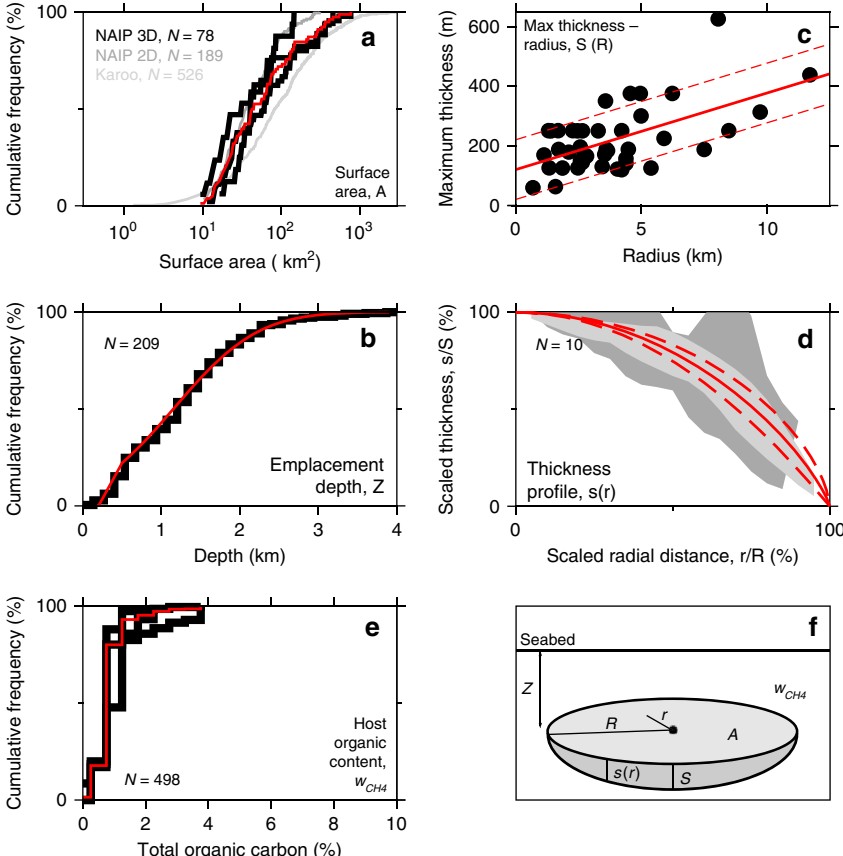

**Fig. 4** Sill dimension observations used for stochastic modelling. On all panels, black lines and grey areas show observations, red lines show distributions used in stochastic modelling. **a** Cumulative distribution of sill surface area. Black lines show observations from three 3D surveys close to the centre of the NAIP, and therefore most likely responsible for peak emissions: Faroe-Shetland Basin (excluding Judd sub-basin), Judd basin and NE Rockall basin. Dark-grey line shows observations from 2D survey compilation from Rockall Basin. Light-grey line shows observations from satellite imaging of Karoo sills (for reference; excluded from definition of red line used for stochastic modelling here). **b** Cumulative distribution of sill emplacement depth, measured from sills imaged on 2D and 3D seismic data across the NAIP basins and around the world. **c** Maximum thickness versus diameter for sills for which clear reflections from both top and bottom of the sill are observed on seismic data. **d** Profiles of scaled sill thickness versus scaled radial distance. Grey envelopes represent mean ± 1 standard deviation of the stacked scaled profiles: dark grey shows sill thickness measured directly; light grey shows sill thickness inferred from forced folds. **e** Host-rock total organic carbon (TOC) content from 498 samples from the Vøring Basin[33]. The three black lines are the distributions for Paleocene, Upper Cretaceous and Lower Cretaceous rocks. **f** Definition of sill measurement terms. See Supplementary Fig. 2 for an example of how sills are observed and measured on seismic reflection data.

that generated the Paleocene–Eocene NAIP, known as the Icelandic mantle plume, can be observed and measured beneath the North Atlantic today. Modern and ancient NAIP basaltic magmas are generated by decompression melting of unusually hot mantle within the head of the Icelandic mantle plume. Flow of mantle rock within this and other plumes is unsteady, or pulsing[34,35]. In the latest Paleocene, a blob of unusually hot mantle was carried up the stem of the plume, between Greenland and Scotland, and spread outwards beneath the North Atlantic plates[36,37]. This hotter mantle formed a ring that travelled rapidly beneath the NW European continental margin[38]. Decompression melting occurred when the hot ring passed beneath the channel of thin lithosphere[39] comprising the Vøring, Møre, Faroe-Shetland and Rockall sedimentary basins, which had formed by rifting during Mesozoic time. Separation of basaltic magma from its mantle source is an efficient process, and about one third of the magma reached the upper few kilometres of the overlying sedimentary basin fill, to be intruded as sills or erupted as lavas[28,29]. These scenarios for NAIP mantle plume pulsing, magma generation and magma transport are generally accepted because North Atlantic oceanic crust contains a continuous record of Iceland plume pulsing between the Paleocene and present[34,35,40]. Up to ten

thermal pulses have been identified through Cenozoic time, and the pulse that generated the NAIP sill province was the largest[34,40].

The mantle plume flux parameter is a fundamental measure of how fast temperature pulses move round a mantle convection cell. It can be directly linked to rates of mantle melting, and here we exploit this opportunity to estimate sill injection frequency. We express the sill intrusion recurrence period as $1/\tau_{\text{repeat}} = Q\rho$, where $Q$ is the rate of expansion of sill province area and $\rho$ is the number of sills per unit area of the seafloor (Methods). We postulate that the area of sill intrusion in the uppermost lithosphere $A_{\text{LIP}}$ is effectively the same as the area of magma generation at the base of the lithosphere, $A_{\text{mantle}}$. This assumption is reasonable because basaltic magma can travel up through the plate to the surface on a timeframe comparable with PETM onset, or shorter[41,42]. If melting occurs across the entire footprint of the expanding patch of unusually hot mantle, then

$$Q = \frac{\mathrm{d}A_{\text{mantle}}}{\mathrm{d}t} = \frac{\mathrm{d}A_{\text{LIP}}}{\mathrm{d}t}. \qquad (2)$$

Thus, $Q$ can also be interpreted as the mantle plume area flux, the rate of areal expansion averaged vertically across the plume head.

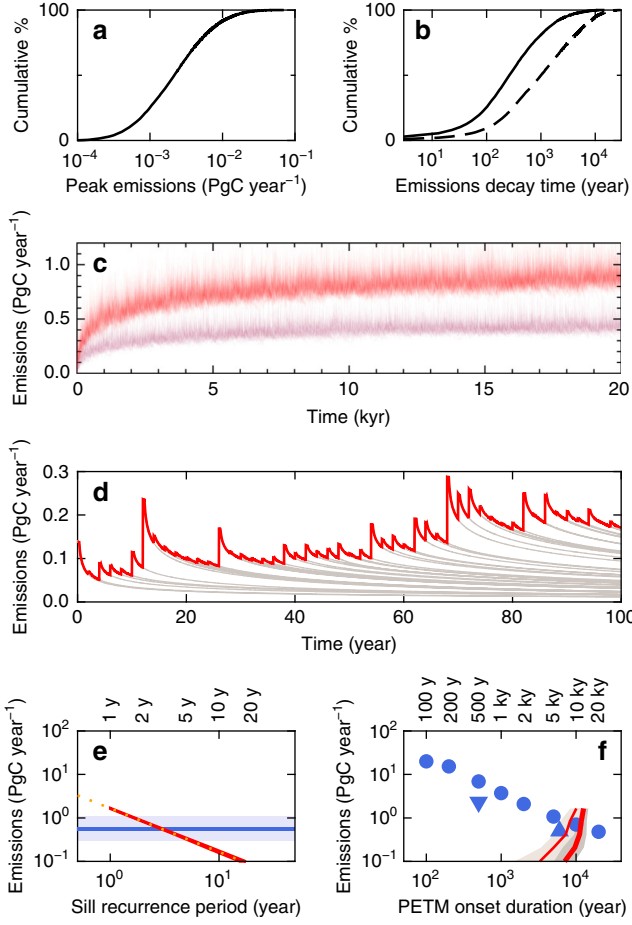

**Fig. 5** Sill province carbon emissions at constant time period between successive sill intrusions. **a** Cumulative histogram of peak carbon emissions (averaged over the first year after injection) for individual NAIP sills, calculated from new NAIP sill dimension database (Fig. 4) using new emissions parameterisation (Methods). **b** Cumulative histogram of emissions decay time for individual NAIP sills calculated in the same way: dashed line for maximum sill thicknesses; solid line for mean sill thicknesses. **c** Stack of 30 simulated carbon emissions histories for constant sill intrusion intervals of $\tau_{repeat} = 2$ years (red) and 4 years (maroon). **d** Expanded view of first 100 years of one simulation with $\tau_{repeat} = 2$ years, showing combined emissions from the sill province as a red line and contributions from individual sills as grey lines. **e** Relationship between long-time (post-10 kyr) mean emissions flux and sill recurrence period from multiple experiments like **c**. Horizontal blue line and envelope show most likely value[5,6] and range[4,5] of peak emissions required to explain PETM onset determined from climate models. Yellow dots show power law approximation in Eq. 1. **f** Comparison between peak carbon fluxes from our sill province models (lines) and peak fluxes estimated from PETM climate models (spots) as functions of PETM onset duration. Sill province models: thinner black line shows when the long-time mean is first reached; thicker black line shows when the 1 kyr running mean first reaches long-time mean; shaded regions show ± 1 s.d. for the 100-run ensembles. Climate models: circles are from joint inverse modelling of pH and carbon isotope composition[7]; triangles are from forward modelling of carbon isotope composition and deep-sea carbonate dissolution (triangle[6]; inverted triangle[57]).

More generally, expansion of the sill province is smaller than $Q$ because thicker lithosphere inhibits decompression melting in some places, so that $A_{LIP} < A_{mantle}$ (Fig. 1). Incorporating this effect modifies the expression for $\tau_{repeat}$ to

$$\frac{1}{\tau_{repeat}} = \rho(R)\Gamma(R)Q,\qquad(3)$$

which is similar to the earlier expression but for addition of a

dimensionless function $\Gamma$ that defines the relationship between $A_{LIP}$ and $A_{mantle}$. Both the sill distribution function $\rho(R)$ and the sill province area function $\Gamma(R)$ can be measured from maps of the sill province based on seismic reflection data.

**Mantle plume flux observations**. Plume area flux $Q$ at the time of the Paleocene/Eocene boundary has been estimated from the relative timing and amplitude of peak dynamic support (i.e. mantle convectively supported uplift) in two sedimentary basins separated by 400 km either side of Scotland: the Judd region of the Faroe-Shetland Basin and the Bressay region of the North Sea[37,38,43]. Dynamic support is a direct indicator of the thermal anomaly that generated the sill province magma. These studies estimate a parameter $k$, with units of diffusivity, which characterises the speed of the locus of peak dynamic support as it travels along a line between the two sedimentary basins. Both the location of the plume centre and the shape of the plume head must be known to convert $k$ to plume area flux $Q$. The published $Q$ values assume a radially symmetrical plume head[37,38,43]. An elliptical plume swell model fits regional dynamic support data better[18,21,44]. Modern dynamic support data suggest that real plume heads are more irregular in planform[45]. To quantify the resulting uncertainty in plume area flux, we re-calculated $Q$ values from published $k$ values[37] using eight different model plume centre locations and their associated plume head aspect ratios (Table 1; Fig. 6). The median value $Q$ value is 4 km² yr⁻¹, the interquartile range is 2.4–6 km² yr⁻¹ and the 10–90 centile range is 1.6–8 km² yr⁻¹. This uncertainty could be significantly reduced in future if more high-quality dynamic support or mantle temperature records can be obtained.

**NAIP sill province geography**. Very few studies state values of $\rho$ (the number of sills per unit area of the seafloor) or provide enough information to derive $\rho$ with confidence. We therefore used seismic reflection data to measure $\rho$ across the NAIP (Table 2). Measurements of $\rho$ from 3D seismic data are most reliable, but relatively few are available. We therefore developed a Monte Carlo approach to correct $\rho$ measurements from 2D data grids for the "missing sills" that exist in the gaps between the 2D lines (Supplementary Fig. 3). We observe 0.07 sills km⁻² near the centre of the NAIP (Fig. 7). For this first attempt at simulating NAIP emissions, we captured the observed decay of $\rho$ with radial distance using a Gaussian of half-width 600 km.

The heavily intruded part of the NAIP sill province comprises the Vøring, Møre, Faroe-Shetland and Rockall chain of sedimentary basins. We measured the area of this basin chain and hence calculated the function $\Gamma(R)$, which captures the relationship between plume head and sill province areas (Fig. 6c). $\Gamma$ initially increases as the leading edge of the plume head arrives beneath the sill province and then decreases roughly linearly with radius.

**Time-dependent model of NAIP carbon emissions flux**. We derived a time-series of variation in $\tau_{repeat}$ from the measurements of plume flux and sill province geography (Fig. 7b). $\tau_{repeat}$ drops to a minimum of 4.6 yr at a time of 8 kyr after sill province initiation. By comparison with the constant-$\tau_{repeat}$ NAIP emissions simulations, it therefore appears possible that the NAIP sill province could have initiated the PETM. We proceeded to use the $\tau_{repeat}$ time-series to estimate how NAIP emissions evolved during the PETM onset period and through the main body of the PETM (Fig. 8). Supplementary Movie 1 shows a Monte Carlo simulation of sills intruding the Vøring-Møre–Faroe-Sheltand–Rockall basin chain at increasing distances from the plume centre as the hot mantle ring beneath sweeps outwards beneath the lithosphere. At

**Table 1 Mantle plume flux data.**

| Plume model | $r_B$ (km) | $r_J$ (km) | $k_{min}$ (km² yr⁻¹) | $k_{max}$ (km² yr⁻¹) | Aspect ratio | Azimuth difference | Geometry factor | $Q_{min}$ (km² yr⁻¹) | $Q_{max}$ (km² yr⁻¹) |
|---|---|---|---|---|---|---|---|---|---|
| WM89 ([28]) | 820 | 580 | 0.3 | 1.7 | 1 | – | 1 | 0.5 | 3.5 |
| LM94 ([56]) | 1260 | 1070 | 0.3 | 2.2 | 1 | – | 1 | 0.7 | 4.6 |
| JW03 ([44]) | 830 | 780 | 0.1 | 0.4 | 1 | – | 1 | 0.1 | 0.8 |
| MJ06g ([18]) | 1000 | 670 | 0.4 | 2.8 | 0.42 | 042 | 1.3 | 1.2 | 7.5 |
| MJ06v ([18]) | 980 | 640 | 0.4 | 2.7 | 0.43 | 054 | 1.7 | 1.5 | 9.7 |
| Nea09 ([21]) | 870 | 540 | 0.4 | 2.3 | 0.36 | 074 | 2.6 | 2.0 | 12.7 |

Plume model provides reference. $r_B$ and $r_J$ are the distances between the model plume centre and the Bressay and Judd sedimentary successions, respectively. The mantle thermal anomaly must have passed Judd before 56.1–55.0 Ma (ref. [36]) and Bressay before 55.8–54.8 Ma (ref. [38]), implying a time difference of $\Delta t$ between 0.2 and 1.3 Myr between passage of the thermal anomaly beneath Judd and Bressay[37]. Hence, plume head parameter $k$ is estimated using $(r_B − r_J)/\Delta t$ (refs. [37,38]). Aspect ratio $\gamma$ (i.e. the ratio of the short and long axes of the ellipse) is specified by the plume model. Azimuth difference $\theta$ is the mean of the differences between the forward azimuth from the plume centre to the Judd or Bressay sedimentary records and the azimuth of the long axis of elliptical plume heads. The geometry factor, given by $(\gamma^2 \cos^2\theta + \sin^2\theta)/\gamma$, corrects the $k$ value, which implicitly assumes radially symmetrical plume spreading, for an elliptical plume head geometry. $Q$ is the plume head area flux, averaged vertically across the plume head, determined using $2\pi k\gamma/3$ (ref. [38]). It is straightforward to convert $Q$ to plume volume flux, mass flux or buoyancy flux[27,38,40]. However, area flux is more directly obtained from the stratigraphical data, and can be more directly related to sill intrusion frequency than these other flux measures

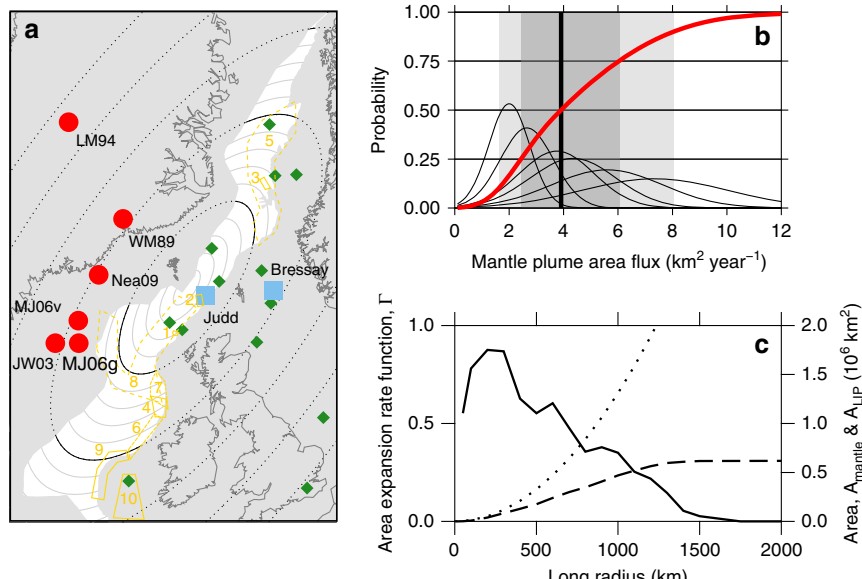

**Fig. 6** NAIP geography and plume flux data. **a** Data locations. Blue squares show locations of stratigraphic successions and river profiles used to constrain plume flux[36,38,43]. Red circles show plume centre locations (references in Table 1). Yellow polygons show seismic datasets used to measure sill area density, $\rho$ (key to numbers in Table 2). Ellipses mark edge of the model plume head used for Figs. 7 and 8 at long radii $R$ in increments of 100 km. Green diamonds show locations of dynamic support measurements from stratigraphic sections used to constrain plume head model[18,21,44]. **b** Mantle plume area flux data. Thin black lines show probability density functions associated with the plume flux estimates in Table 1, assuming normal distributions with the standard deviations set to one-quarter of the uncertainty ranges. Red line: corresponding cumulative probability function. Thick black line marks median area flux; darker grey box shows inter-quartile range of area flux; lighter grey box shows 10–90% confidence interval for plume area flux. **c** Measurement of NAIP sill province area expansion. $A_{mantle}$ (dotted) is the total area inside an ellipse in **a**. $A_{LIP}$ (dashed) is the area of the NAIP sill province (coloured white) within the ellipse. $\Gamma$ (solid) is the areal expansion rate function.

the median estimated mantle plume flux of $4\,\mathrm{km^2\,yr^{-1}}$, peak carbon emissions flux of $0.37 \pm 0.05\,\mathrm{PgC\,yr^{-1}}$ occurs $11.7 \pm 4.1\,\mathrm{kyr}$ after province initiation, and a cumulative mass of $2750 \pm 1280\,\mathrm{PgC}$ was delivered by this time (means of 100 stochastic runs ± 1 s.d.). For the interquartile range of mantle plume flux, the peak emissions flux range is $0.2–0.5\,\mathrm{PgC\,yr^{-1}}$ and occurs at $9–15\,\mathrm{kyr}$ after province initiation. The modelled rise in carbon emissions flux is convex upwards for all values of mantle plume flux, with over half of eventual peak flux reached $<5\,\mathrm{kyr}$ after province initiation.

Emissions flux histories independently inferred from records of oceanic carbonate chemistry[4–6] indicate peak emissions flux of $0.3–1.1\,\mathrm{PgC\,yr^{-1}}$ occurred $<20\,\mathrm{kyr}$ after initiation[5,8–10], with cumulative emissions to peak of $1800–4700\,\mathrm{PgC}$ (refs. [5,6]). Based on this first ever direct comparison of the carbon source and sink,

we conclude that it is possible, indeed likely, that predominantly thermogenic carbon emissions associated with the NAIP sill province was the principal driver of the PETM global climate change event.

## Discussion

Our mantle process approach to the NAIP duration problem shows that greenhouse gas emissions could have occurred rapidly enough to initiate PETM climate change. The initial rapid emissions increase is a consequence of high flow speeds within the solid asthenospheric mantle, because rapid delivery of hot mantle source into the melting zone leads to high magma production rates. Our model has lateral mantle flow rates of over $400\,\mathrm{mm\,yr^{-1}}$ near the NAIP sill province centre, which are determined from measurements of temporally changing surface

**Table 2 Sill intrusion density observations.**

| Basin location | ID | Survey name | Distance (km) | Area (km²) | $N_{obs}$ | $N_{corr}$ | Area density, $\rho$ ($10^{-3}$ sills km$^{-2}$) |
|---|---|---|---|---|---|---|---|
| Rockall | 1 | NEROCK | 280 ± 50 | 493 | 36 | – | 73 |
| Faroe-Shetland | 2 | Judd | 200 ± 80 | 1330 | 62 | – | 47 |
| Møre ([47]) | 3 | Edvarda | 630 ± 20 | 1440 | 27 | – | 30 |
| Rockall | 4 | INROCK | 800 ± 100 | 8077 | 98 | 290–340 | 39 ± 3 |
| Vøring & Møre ([17]) | 5 | Various | 750 ± 250 | 80,000 | 735 | 2205–3675 | 37 ± 9 |
| Rockall | 6 | DGER96 | 850 ± 150 | 38,500 | 191 | 545–610 | 15 ± 1 |
| Rockall | 7 | DGWH96 | 650 ± 150 | 19,090 | 85 | 190–230 | 11 ± 1 |
| Rockall | 8 | OGA | 475 ± 325 | 145,000 | 920 | 2786–2970 | 20 ± 1 |
| Rockall | 9 | ISROCK | 1100 ± 200 | 41,167 | 46 | 108–139 | 3 ± 0.5 |
| Porcupine | 10 | Various | 1500 ± 300 | 75,868 | 112 | 240–310 | 4 ± 0.5 |

All area density data based on our own seismic interpretations except where a reference number is given next to the basin location. ID numbers show location on Fig. 6. Distance is relative to the plume centre in Figs. 6 and 8. $N_{obs}$ is the number of sills observed on the seismic data. $N_{corr}$ is the number of sills after correction for missing sills on 2D surveys (Supplementary Fig. 4); a dash in this column indicates a 3D survey that does not require a missing sills correction

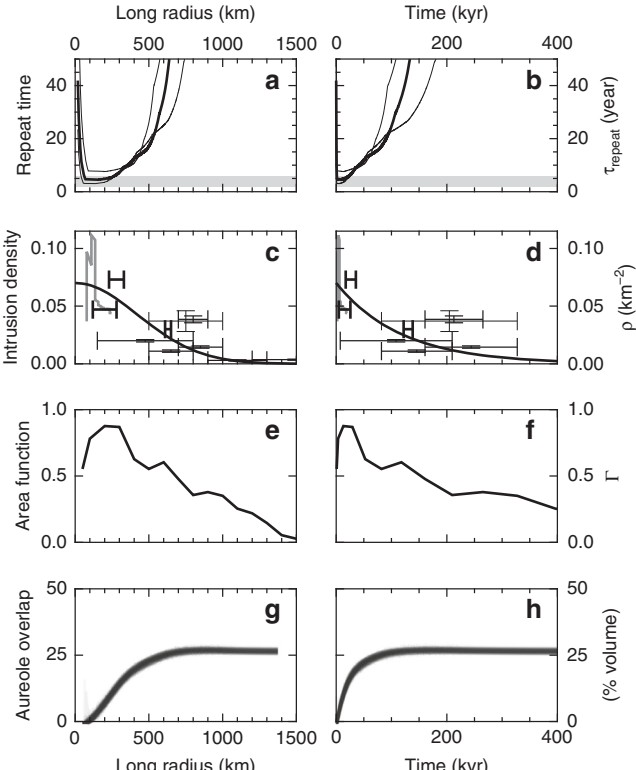

**Fig. 7** Solid earth controls on onset and decay of carbon emissions. **a**, **b** Time between successive sill intrusions ($\tau_{repeat}$) as a function of long radius of the underlying mantle thermal anomaly (**a**) and of time (**b**), used for the simulations in Fig. 8. Thick and thin lines show median and interquartile range of plume flux estimates. Horizontal grey bar shows $\tau_{repeat}$ range required to explain PETM onset, based on the constant-$\tau_{repeat}$ experiments (Fig. 5). **c**, **d** Crosses show observed number of sills per unit area, $\rho$, measured on seismic reflection data (thicker lines from 3D data; thinner lines from 2D data); vertical error bars show total uncertainty range in $\rho$ (Table 2); horizontal error bars show range in radius/time corresponding to footprint of seismic survey (Fig. 6). Black curve shows smooth parameterisation of the observations, used in Fig. 8. Grey line shows alternative $\rho$ distribution inferred from a carbon sink-derived emissions estimate[5] (Methods; Supplementary Fig. 5). **e**, **f** Area function $\Gamma$ that relates expansion rates of the mantle thermal anomaly (magma source) and the NAIP sill province (magma fate). Functions $\Gamma$ ($r$) and $\rho(r)$ control sill intrusion period according to $1/\tau_{repeat} = Q\,\Gamma(R)\,\rho(R)$ (see Results for derivation), where $Q$ is the mantle plume area flux and $R$ is the long radius of the plume head. **g**, **h** Cumulative proportion of overlapping thermal aureoles (stack of 100 stochastic runs).

uplift observed within the stratigraphic record[36–38,43]. For comparison, asthenospheric mantle flow speeds of 200–300 mm yr$^{-1}$ are measured south of Iceland at present and through the Neogene[34,35], and it is unsurprising that flow rates during the Paleocene–Eocene transient LIP initiation phase should be higher[28].

Our simulations are also characterised by a rapid post-peak decline in emissions flux. This decrease arises from a combination of three effects (Fig. 8f). First, the number of sills in a given area decreases with distance from the province centre. This observation can be explained because the magnitude of a radially spreading mantle thermal anomaly decreases with distance[35,38]. Secondly, the proportion of the anomalously hot mantle patch that is overlain by the sill province also decreases with radial distance. Both these geographic functions are initially at or close to maximum, and decrease most rapidly over the first 50,000 years, leading to increasing $\tau_{repeat}$ (Fig. 7). Thirdly, as more sills are intruded, new sills are increasingly likely to intrude an existing thermal contact aureole, so that otherwise similar sills intruding at later times tend to generate less methane.

Over 90% of the carbon emissions from the sill province come from thermogenic methane, with the remainder from mantle-derived carbon dioxide (Fig. 8e). Thermogenic methane contributes most carbon for two reasons. First, sill magma typically contains 0.5% $CO_2$ by mass, or 0.14% carbon, whereas the mean organic carbon content within NAIP sill host rocks frequently exceeds 1% (Fig. 4e). Secondly, the width of the thermal aureole is between 2 and 8 times the sill thickness (Supplementary Fig. 3a, b).

We focus here on comparing our emissions flux histories with recent inverse modelling of surface ocean pH records[5] and forward models of deep-sea dissolution[6]. Such records provide tighter constraints on carbon emissions flux than records of carbon isotopic composition alone because of trade-off between the mass and isotopic composition of the emitted carbon. Nevertheless, a large literature on carbon isotopic records across the PETM exists. We therefore estimated the carbon isotopic composition of NAIP sill province methane in order to demonstrate compatibility with PETM C-isotope records as well as the newer pH and carbonate dissolution records (Fig. 8d; Supplementary Fig. 4). The predicted carbon isotopic composition of NAIP sill province emissions is near the limit of the uncertainty range of isotopic compositions estimated by joint inverse modelling of isotopic composition and surface ocean pH, but closely comparable with the estimated composition for a sensitivity experiment with the same PETM onset duration as our simulation[5] (Fig. 8d). More generally, our models can easily explain the globally observed negative shift of several per mil. over the PETM onset period[7].

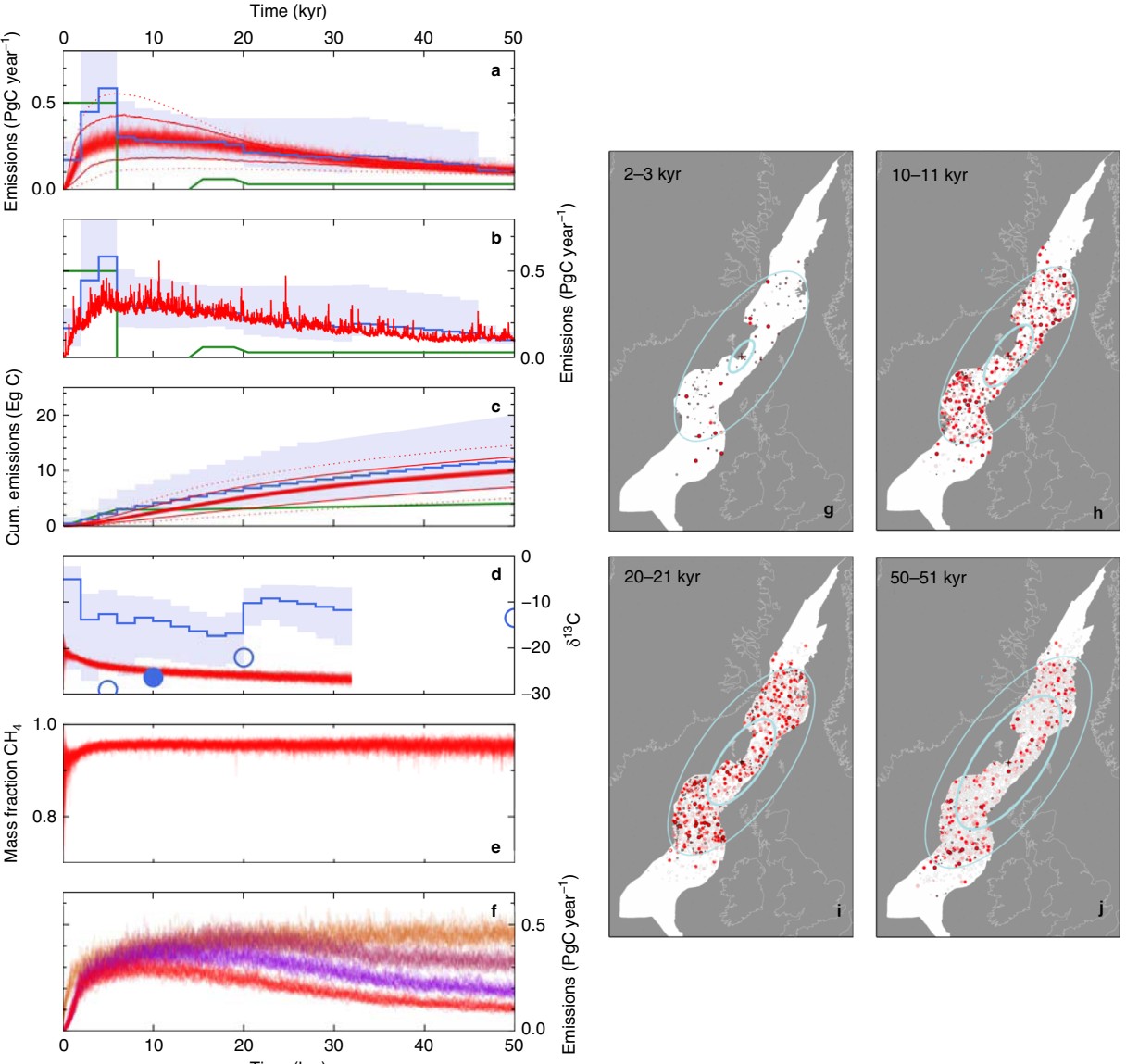

**Fig. 8** NAIP carbon emissions based on mantle convection model, and comparison with climate-based emissions reconstructions. **a** The red cloud is a stack of 100 simulated NAIP emissions flux histories with variable $\tau_{repeat}$ controlled by plume head model and NAIP geography and the median mantle plume flux estimate. Red lines show ensemble means using the upper and lower quartile mantle plume flux estimates; dotted red lines show ensemble means using the 10th and 90th percentile plume flux estimates. The blue line and envelope show the emissions estimate and uncertainty range from joint inverse modelling of oceanic pH and carbon isotope records[5]. The green line shows emissions estimated from forward modelling of carbon isotope and deep-sea carbonate dissolution records[6,13]. **b** Emissions flux for one stochastic run. **c** Cumulative emissions associated with a. **d** Isotopic composition of emissions. Blue circles are sensitivity experiments that show impact of uncertainty in PETM onset duration on mean isotopic composition of emissions[5]; filled circle most closely comparable with our simulations (10 kyr onset). Blue line/envelope truncated at 30 kyr, when significant carbon burial begins to influence predicted composition of emissions[5]. **e** Proportion of total emissions released as thermogenic methane. **f** Exploration of mechanisms for post-peak reduction in emissions itemised in Fig. 7: red stack, same as **a**; purple stack, ignoring effect of overlapping sill aureoles; maroon stack, ignoring effects of overlapping aureoles and radial decrease in sill intrusion density; brown stack, constant $\tau_{repeat} = 4$ yr, i.e. ignoring effects of overlapping aureoles, radial decrease in sill intrusion density and radial spreading of magma generation zone. **g–j** Snapshots from one NAIP simulation (Supplementary Movie 1) showing sills intruding the the Rockall–Faroe-Shetland–Møre-Vøring sedimentary basin chain at progressively increasing distances from province centre. Red circles show sill–vent systems currently emitting carbon: the shade of red is proportional to the emissions flux and the size is proportional to final cumulative emissions (horizontal scale exaggerated). Older sill–vent systems no longer emitting carbon are coloured grey. Pale blue ellipses show positions of the thermal anomaly in the sub-plate convecting mantle, which generates the sill-province magma: the heavy line marks the peak thermal anomaly; the thin line indicates 5% of the peak thermal anomaly.

We only account here for mantle-derived carbon released from shallowly intruded NAIP sill magma, generated by lithospheric thin-spot melting (Fig. 1). Magma generated within the Europe-Greenland continental break-up zone released further mantle-derived carbon. The total volume of lavas and deep intrusions comprising the NAIP magma-rich passive margins is $3.8 \times 10^6$ km$^3$ (ref. [29]), which could release 14,100 PgC of carbon with a mantle-derived isotopic signature. The duration of eruption/emplacement, related to the duration of rifting leading to continental breakup, is 1–3 Myr (refs. [28,29]), considerably greater

than the timeframe of emplacement of the sill province. The resulting emissions flux of 0.005–0.015 PgC yr$^{-1}$ and mass of 50–150 PgC from lavas and deep intrusions during the PETM onset period is negligible in comparison with the flux of 0.2–0.5 PgC yr$^{-1}$ and mass of 2750 PgC from thermogenic methane and sill-degassed carbon dioxide. This difference in emissions flux reflects the difference between the relatively low rates of mantle upwelling and melting driven by plate spreading, and the significantly greater mantle upwelling and melting rates driven by convective flow close to the centre of a major plume head.

Following most others, we have assumed that all generated methane can escape the solid Earth to the ocean–atmosphere system[13,17]. However, the proportion of generated to emitted methane is not well known, and neither is it clear whether escaping methane is emitted at the rate it is generated, or at a slower rate[46]. Field and seismic observations of multiple vents per sill[47], ubiquitous occurrence of fracture networks next to igneous sheets, and the depth range of venting (Fig. 4) show that sill-induced hydro-fracking is common in the upper few kilometres of sedimentary basins. Such observations underlie the widespread assumption that generated methane escapes efficiently. On the other hand, a model of the sill-associated hydrothermal convection system suggests the proportion of methane that escapes may be lower[46,48]. Since it is clear that shallowly intruding sills generate their own fracture permeability, any hydrothermal system model must include a description of how the fracture permeability varies with pressure. This relationship is controlled by the compressibilities of the sill magma/rock and host rock[49], which have not yet been factored into models of hydrothermal systems adjacent to igneous sills[46,48]. The relationship between generated and emitted methane is central to predicting environmental consequences, and reliable models to assess it are urgently required.

We have presented the first mechanistic model of carbon emissions flux from any proposed PETM carbon source that is directly constrained by measurements of the geological structures that control the emissions. Differences between our carbon emissions source function and carbon emissions inferred independently from climatic effects provide for the first time the potential to directly measure climate system feedbacks. However, it is too early to interpret source-sink differences in Fig. 8 in terms of such feedbacks. For example, modelling the carbon sink[5,6] indicates peak emissions flux of 0.5–0.6 PgC yr$^{-1}$ followed by a sharp decrease. Our median estimate of the carbon emissions source has a peak 0.2–0.3 PgC yr$^{-1}$ lower and a smoother post-peak decrease. However, the smoothness of our carbon flux estimate derives partly from the smooth curve used to represent the sill distribution ($\rho$) measurements (Fig. 7c). Given sparse $\rho$ measurements near the sill province centre, we cannot yet rule out a carbon emissions source function that would match emissions inferred from inverse modelling of the carbon sink more closely (Supplementary Fig. 5). Considering uncertainties in the other controls on $\tau_{repeat}$, peak emissions flux could occur between 1 and 20 kyr (Figs. 5 and 8). Thus, more sill distribution measurements, more mantle plume flux measurements and improved measurements of the shape and positioning of the plume head are required to reduce the uncertainty range of our solid Earth emissions model. These combined uncertainties at present leave room for a scenario in which no significant greenhouse gas sources external to the NAIP sill province are required to explain the PETM onset and the first few tens of thousands of years of the PETM recovery. Clarification of this carbon cycle behaviour will impact modelling and management of future climate change. We therefore eagerly anticipate improvements in modelling both the source and fate of carbon for the NAIP-PETM pair, and also other LIPs paired with major climate change events.

## Methods

**Kinetic reaction modelling of thermogenic emissions.** The aims are to determine time-series of carbon emissions flux for comparison with PETM climate records (previous work reports final total mass of emissions[17,25]), to cover the full observed range of sill thicknesses and emplacement depths (previous work covers thinner, shallower sills[25]), and to calculate methane generation directly (previous work uses vitrinite reflectance as a proxy[25,50]). We combined several established methods to achieve these aims. All calculations were done in 1 spatial dimension: depth in the case of a sill, or horizontal distance in the case of a dyke. 1D calculations were used because the aim was to produce a simple parameterisation flexible enough to cover the observed range of sill and host-rock characteristics yet fast enough to be run very many times for stochastic modelling. The dyke calculations were used to test the numerical thermal calculations against analytical solutions for dykes, and to test the reaction calculations by comparison with published thermal aureole measurements for dykes.

Thermal maturation calculations were begun many millions of years before igneous intrusion to simulate burial heating and consequent maturation, which has a significant effect on final aureole width after intrusion. Methane already generated by burial maturation at the time of igneous intrusion is not included in post-intrusion emissions. During this stage, temperature is given by

$$T = T_0 + G(z + Bt),  \qquad (4)$$

where $T_0$ is the seabed temperature, $G$ is the geothermal gradient, $B$ is the burial rate and $z$ is depth. A geothermal gradient of $G = 30\,°C\,km^{-1}$, burial rate of $B = 10^{-5}\,m\,yr^{-1}$ and seabed temperature of $T_0 = 10\,°C$ were used for all calculations (all notation given in Supplementary Table 2). This $G$ value may underestimate the true value in the Vøring and Møre Basins[51] but may overestimate the true value in the Rockall Basin, where extreme crustal extension has occurred.

Instantaneous sill intrusion was assumed[33]. Thermal evolution was then tracked by accounting for conductive cooling of the igneous sheet and host rock according to

$$\frac{\partial T}{\partial t} = \kappa \frac{\partial^2 T}{\partial z^2},  \qquad (5)$$

where $\kappa$ is the thermal diffusivity. Before solidification, this equation was solved using the method given in §4.19 of ref. [27], which accounts for latent heat of crystallisation. After solidification, Eq. 4 was solved using an FTCS explicit finite difference scheme. A uniform distance grid was used, and the grid spacing was kept small enough to generate smooth emissions flux time-series: typically 1 m up to 250 years post intrusion, 5 m between 250 and 10,000 years after intrusion, and 20 m at greater times. Constant heat flux was held fixed. Seabed temperature was assumed at the boundary/ies far from the sill/dyke, and the distance grid was made long enough to minimise edge effects. Burial heating after intrusion was not considered because only times of <1 Myr after intrusion are of interest here, during which burial heating is negligible in comparison with the thermal effect of the igneous sheet. At times well after complete solidification, an analytical solution is available to check these calculations (ref. [27], §4.21).

Four thermal maturation reaction groups were tracked[31,32]. The first group, RG1, describes thermal maturation of labile (oil-prone) kerogen to produce mostly oil and a smaller amount of methane. The second group, RG2, describes thermal maturation of refractory (gas-prone) kerogen to produce methane. The third group, RG3, describes thermal maturation of vitrinite, leading to increasing reflectance. The fourth group, RG4, describes cracking of oil (produced by RG1) to produce methane. Each reaction group covers a natural range of kerogen composition, with a corresponding range of activation energies. Therefore, each group is described by three parameters: a frequency factor $A_E$, a mean activation energy $E_a$ and a standard deviation of activation energies $\sigma$ (Supplementary Table 2).

Our reaction modelling implements the kinetic models of refs. [31] and [32]. We approximated each normal distribution of activation energies with an odd number $N_E$ of discrete reactions, each with a constant activation energy and frequency factor. This strategy is more straightforward to implement and quicker to execute than numerical integration because progress of each reaction at each time step can be calculated using the analytical expressions. The concentration of a product $P$ (i.e. the progress of the reaction) for one of the discrete reactions in RG1 to RG3 is given by

$$\frac{d(1 - P)}{dt} = -\bar{k}(1 - P),  \qquad (6)$$

where $\bar{k}$ is the average reaction rate. We assume $\bar{k}$ to be constant within each short time step of the finite difference calculation; the harmonic mean of the reaction rates for the temperatures at the start and end of the time step was used. The reaction rate at any temperature is given by the Arrhenius equation

$$k = A_E \exp\left(-\frac{\overline{E_a}}{R_g T}\right),  \qquad (7)$$

where $A_E$ is the frequency factor, $\overline{E_a}$ is the mean activation energy of the discrete reaction and $R_g$ is the gas constant. The reaction progress at the end of the time step is then given by

$$P = 1 - (1 - P_0)\exp(-\bar{k}t),  \qquad (8)$$

where $P_0$ is the progress at the start of the time step. We assumed that the total

amount of kerogen was initially evenly distributed over the discrete reaction in each group, so that overall progress of the reactions group at each time step is given by

$$P = \sum_{i=1}^{N_E} \frac{P_i}{N_E} . \tag{9}$$

The discrete oil cracking reactions in RG4 are governed by

$$\frac{d(1-P)}{dt} = -\bar{k}(1-P) - C , \tag{10}$$

where $C(t)$ is the total rate of oil generation from the RG1 reactions. We assume $C$ to be constant over each short time step so that

$$P = 1 - \frac{C}{\bar{k}} - \left(1 - P_0 - \frac{C}{\bar{k}}\right) \exp(-\bar{k}t) . \tag{11}$$

We also assume that the oil generated at each time step is distributed evenly over the discrete oil cracking reactions, so that the terms involving $C$ become $C/\bar{k}N_E$. The individual oil cracking reactions were then summed as before to find the reaction progress for the group.

A discrete reaction within any reaction group covers a range of activation energies between $E_0$ and $E_1$ given by

$$\int_{E_0}^{E_1} \Pi(E)dE = \frac{1}{N_E} , \tag{12}$$

where $\Pi(E)$ is the normal probability density function. Substituting this function and integrating gives

$$E_1 = E_a - \sigma\sqrt{2}\,\text{erf}^{-1}\left[\text{erf}\left(\frac{E_a-E_0}{\sigma\sqrt{2}}\right) - \frac{2}{N_E}\right] . \tag{13}$$

Thus, the values $E_0$ and $E_1$ that bound each of the $N$ chunks of the overall activation energy distribution can be found by stepping across the distribution. The mean activation energy for each discrete reaction is then given by

$$\overline{E_a} = N_E \int_{E_0}^{E_1} E\,\Pi(E)dE , \tag{14}$$

which evaluates to

$$\overline{E_a} = N_E \left[\frac{-\sigma}{\sqrt{2\pi}}\exp\left(\frac{-(E_a-E)^2}{\sigma\sqrt{2}}\right) - \frac{E_a}{2}\text{erf}\left(\frac{E_a-E}{\sigma\sqrt{2}}\right)\right]_{E_0}^{E_1} . \tag{15}$$

The method developed here is a simple way of achieving our aims of directly estimating time-series of methane emissions flux over the observed range of individual sill dimensions. Additional effects such as latent heat of mineral organic maturation[25], mineral dehydration reactions[25] and detailed thermal characteristics of the host rock[50] are included in other schemes for estimating thermogenic methane generation. Our method could be augmented to include these effects, though we anticipate that two other effects are more significant and will require more urgent attention. First, there is uncertainty in the proportion of generated methane released from the solid Earth (Discussion). Secondly, our NAIP sill province simulations show that 25% of sill aureoles interact with each other (Fig. 7g, h), implying that a wider range initial conditions for thermal structure and organic reaction progress should be covered in future calculations.

**Parameterisation of thermogenic emissions results**. We sought an analytically solvable expression with minimum free parameters for use in stochastic modelling. We found that the cumulative mass of carbon $m_{therm}(\Delta t)$ generated by thermal maturation adjacent to an igneous sheet can be conveniently parameterised as

$$m_{therm}(\Delta t) = M_{therm}\left(1 - \exp\left[-\left(\frac{\Delta t}{\tau_{decay}}\right)^p\right]\right) , \tag{16}$$

where $\Delta t$ is the time since intrusion, $M_{therm}$ is the total mass of carbon released by thermal maturation when the sheet has fully cooled and $\tau_{decay}$ is the characteristic decay time for gas release. The flux of thermogenic methane is then found by differentiating this expression to give

$$q_{therm}(\Delta t) = p\left(\frac{M_{therm}}{\tau_{decay}}\right)\left(\frac{\Delta t}{\tau_{decay}}\right)^{p-1}\exp\left[-\frac{\Delta t}{\tau_{decay}}\right] . \tag{17}$$

This parameterisation is empirical but it reflects the real processes operating within the thermal aureole of the sill quite closely. The cumulative mass of carbon generated must increase until it tends to a constant value when the sill has cooled sufficiently to inhibit further maturation reactions. The analytical solution for conductive heating of the host rock shows that the thermal front moves outwards as a power law with $p = 0.5$ (ref. [27]). Initially, we expect the thermogenic reaction front to move out from the sill at a similar rate to the thermal front but with a $p$ exponent slightly >0.5 because reactions can take place at lower temperatures when more time is available. For the same reason, the emissions decay time will be of the same order but slightly larger than the conductive cooling time given by

$$\tau_{cond} = \frac{S^2}{\kappa} , \tag{18}$$

where $S$ is the sill thickness and $\kappa$ is the thermal diffusivity of the host rock. At later times, equivalent to maturation at distances further from the sill, the maximum temperature experienced by the host rock becomes exponentially smaller[27]. Thus, the expressions for $m_{therm}$ and $q_{therm}$ behave as power laws when $\Delta t < \tau_{decay}$ and taper asymptotically to $M_{therm}$ or zero when $\Delta t > \tau_{decay}$.

The singularity at $q_{therm}(\Delta t = 0)$ arises because of the assumption of instantaneous intrusion. This assumption is reasonable because the intrusion

period is on the order of days whereas the gas generation occurs over decades to thousands of years. To avoid the singularity when plotting at time $t \ll 1$ yr and when quoting initial fluxes, we average $q_{therm}$ over a preceding time period $\bar{t}$ using

$$\overline{q_{therm}}(\Delta t, \bar{t}) = \frac{M_{therm}}{\bar{t}}\left(\exp\left[-\left(\frac{[\Delta t - \bar{t}]}{\tau_{decay}}\right)^P\right] - \exp\left[-\left(\frac{\Delta t}{\tau_{decay}}\right)^P\right]\right) . \tag{19}$$

An advantage of the empirical form developed above is that the parameter set $(M_{therm}, \tau_{decay}, p)$ required to specify $m_{therm}$ and $q_{therm}$ is straightforwardly related to the set of parameters $(A, K, S, Z, w_{CH4}, \beta)$ that describes the dimensions of the igneous sheet and the characteristics of the host rock. These are: $A$, the surface area of the sheet; $Z$, the depth of intrusion of the sheet below the contemporary seabed; $S$, the maximum thickness of the sheet; $\beta$, an exponent that describes how the sheet thins towards its edges; $w_{CH4}$, the weight% organic matter expelled as carbon after thermal maturation; and $K$, the proportion labile:(labile + refractory) kerogen, where labile kerogen is the term for organic matter that initially produces oil which then cracks to gas, and refractory kerogen is the term for organic matter that produces gas directly[31].

The total carbon mass parameter is determined using

$$M_{therm} = A\,S\,Y\,d_{host}\,w_{CH4} , \tag{20}$$

where $Y$ is the final width of the thermal maturation reaction aureole scaled in units of maximum sheet thickness $S$, and $d_{host}$ is the density of the host rock. Note that our definition of $Y$ includes the thermogenic reaction aureoles either side of the sheet (but not the igneous sheet itself), whereas some other studies use the scaled thickness of the aureole on one side only[25]. Our definition implicitly accounts for significant differences in thickness of the aureoles above and below a sill (Fig. 2), which arise because the speed of the contact maturation reactions is enhanced by the increase in pre-intrusion burial maturation with depth, and retarded by the fixed sediment surface temperature. These effects give a fourfold variation in $Y$ across the depth range of interest (Supplementary Fig. 1). The functions $Y_{crack}(S, Z)$ and $Y_{ref}(S, Z)$, the scaled reaction aureole widths for methane from oil cracking and refractory kerogen maturation, respectively, were determined from multiple runs of the coupled thermal-kinetic reaction model that spanned the range of observed $S$ and $Z$ values (e.g. Fig. 2). $Y$ was then found by averaging using

$$Y = K\,Y_{crack} + (1 - K)Y_{ref} . \tag{21}$$

Sills that vent fluid from their aureoles are intruded into the shallowest part of the sedimentary column, which is most affected by compaction. Sill emplacement depth therefore affects carbon emissions flux via host-rock density $d_{host}$. The standard relationship between density and porosity is

$$d_{host} = \phi\,d_{water} + (1 - \phi)d_{grain} , \tag{22}$$

where $d_{water} = 1030$ kg m$^{-3}$ and $d_{grain} = 2650$ kg m$^{-3}$ are the densities of pore water and solid rock grains, respectively. The standard exponential porosity–depth relationship is

$$\phi = \phi_0 \exp\left[-\frac{Z}{\lambda}\right] , \tag{23}$$

where we use a depositional porosity of $\phi_0 = 0.61$ and a compaction length scale of $\lambda = 2$ km, both appropriate for mudrocks. These expressions show that the density at the observed mean emplacement depth (1.2 km) is 2110 km m$^{-3}$, which was used for the calculations in Figs. 2, 3 and Supplementary Fig. 1.

The carbon emissions parameters $\tau_{decay}$ and $p$ are simple to relate to the sill dimension parameters. The emissions decay time is expressed as

$$\tau^* = \frac{\tau_{decay}}{\tau_{cond}} . \tag{24}$$

The functions $\tau^*(S, Z)$ and $p(S, Z)$ were determined from multiple runs of the coupled thermal-kinetic reaction model that spanned the range of observed $S$ and $Z$ values (Supplementary Fig. 1). As anticipated, $\tau^*$ values are slightly greater than $\tau_{cond}$ and $p$ values are slightly greater than 0.5.

To account for variation in thickness across each sill within the rapid 1D calculation framework, we assumed radial symmetry in thickness, subdivided each sill into five annuli of equal surface area, calculated $M_{therm}$ and $\tau_{decay}$ using the mean thickness within each annulus, and summed fluxes for each annulus to find the flux from the whole sill.

**Parameterisation of carbon emissions from magma degassing**. The total mass of carbon dissolved in the magma is

$$M_{magma} = A\,S\,d_{magma}\,w_{CO2} , \tag{25}$$

where $d_{magma} = 2750$ kg m$^{-3}$ is the density of the magma and $w_{CO2} = 0.5\% \times 27\%$, the former being the typical wt% $CO_2$ and the latter the amount of carbon in the $CO_2$ molecule. Carbon is strongly incompatible in all primary minerals formed during solidification of basaltic sills. Almost all carbon dioxide dissolved in the magma therefore exsolves shortly before solidification, mixes with the super-critical host-rock pore fluid, and escapes through hydrothermal vents along with the thermogenic methane. The proportion of solidified sill to magma is a roughly linear function of time, so the mass of magmatic carbon exsolved is

$$m_{magma}(\Delta t < \tau_{solid}) = M_{magma}\,t/\tau_{solid} \tag{26}$$

and the flux of magmatic carbon emissions is

$$q_{magma}(\Delta t < \tau_{solid}) = M_{magma}/\tau_{solid} \, , \tag{27}$$

where $\tau_{solid}$ is the time to full solidification, which we parameterised as

$$\tau_{solid} = 0.0147 - 0.000141\,S + 0.00472\,S^2 \, , \tag{28}$$

which gives $\tau_{solid}$ in years for $S$ in metres. After the sill has fully solidified, we have

$$m_{magma}(\Delta t \geq \tau_{solid}) = M_{magma} \, , \tag{29}$$

$$q_{magma}(\Delta t \geq \tau_{solid}) = 0 \, . \tag{30}$$

**Sill and host-rock observation database.** Our primary set of surface area observations comes from several 3D seismic reflection surveys close to the centre of the NAIP. These were checked with reference to 2D data from across the NAIP. Sill lengths measured on 2D surveys (e.g. Supplementary Fig. 2) were converted to areas assuming a circular planform. The sill area histogram for each 2D survey was corrected for the fact that the sill chord intersected by a randomly oriented seismic line rarely represents the true sill diameter. For both 3D and 2D data, we excluded surface areas < 10 km² because hydrothermal vents are rarely observed associated with such small sills, so their methane likely migrates over a timeframe longer than that of interest here. Sill surface areas measured from satellite images of the Karoo sill province are shown for comparison in Fig. 4a but were not included in the surface area distribution used for stochastic modelling.

Depths of intrusion were estimated from the subset of sills for which the coeval seabed can be identified from associated vents and/or forced folds. Measured emplacement depths were backstripped to correct for post-emplacement compaction, using the same porosity model parameters as for the host-rock density calculation.

Maximum sill thickness was measured for the subset of sills that show seismic reflections from both top and base, on both 3D and 2D datasets. The maximum sill thickness used in the stochastic modelling was obtained by first applying the linear regression to the diameter equivalent to the stochastically chosen surface area (assuming a circular sill), and then perturbing this result by a thickness selected from a uniform distribution between ±100 m.

We measured sill thickness as a function of radius both directly, where top and bottom reflections were visible, and indirectly from forced folds in the contemporary seabed. Thickness profiles near sill tips can only be estimated from forced folds because the resolution of oil-industry seismic data prohibits reflections from the bottom of the sill when its thickness is smaller than about 70 m. Fold-derived thickness profiles were corrected for compaction owing to post-intrusion sediment deposition. Other factors such as flexural host-rock deformation, anomalous cementation, fold-crest erosion and interaction with adjacent forced folds could also possibly corrupt sill thickness profiles derived from folds. We minimised influence of these complicating factors by only selecting examples where the thickness profiles from decompacted folds and direct sill imaging agree within measurement uncertainty. Maximum thickness estimated from the forced fold was used to scale the thickness profiles for both the forced fold and the directly imaged sill. The distance between the fold-derived thickness maximum and the fold-derived edge of the sill is then used to scale distance. The two profiles from a sill centre to the tips on either side were scaled separately in terms of distance.

**Stochastic carbon emissions modelling procedure.** The stochastic modelling algorithm has nine steps. The first step is to determine $\tau_{repeat}$, the time elapsed since intrusion of the previous sill: either specify $\tau_{repeat}$ a priori or allow $\tau_{repeat}$ to vary as a function of mantle plume flux, sill intrusion density data and sill province area data. Secondly, define a new sill–vent system by drawing eight random numbers for use in steps 3–5; physical properties thus specified remain fixed for subsequent time steps. Thirdly, determine new sill dimensions: define sill surface area $A$, depth of intrusion $Z$, maximum thickness $S$ and thickness profile exponent $\beta$ using the cumulative probability distributions in Fig. 4. Step 4 is to determine host-rock characteristics for the new sill: define the weight fraction that converts to methane $w_{CH4}$ and the kerogen composition parameter $K$ using the cumulative probability distributions in Fig. 4. Step 5 is to determine thermogenic carbon generation parameters for the new sill: use the new thermogenic carbon emissions para-meterisation to estimate the final mass of generated carbon $M_{therm}(A, K, S, Z, w_{CH4}, \beta)$, the emissions decay time $\tau_{decay}(S, K, Z)$ and the emissions decay exponent $p(S, K, Z)$ from the sill dimensions and host-rock characteristics obtained in steps 3 and 4. Step 6 is to determine new sill location, if required: determine polar coordinates based on the plume head model[31] and known sill province geography, determine aureole overlaps, and correct $M_{therm}$ to avoid double-counting. Step 7 is to determine direct magma degassing carbon emissions parameters for the new sill: the final mass of degassed carbon $M_{magma}(A, S)$ and the emissions decay time $\tau_{solid}(S, Z)$ are estimated from the sill dimensions obtained in step 3. Step 8 is to determine carbon emissions flux and cumulative emitted carbon from the entire sill province by summing thermogenic emissions $(q, m)(M_{therm}, \tau_{decay}, p, t)$ and magma degassing emissions $(q, m)(M_{magma}, \tau_{solid}, t)$ from all sill–vent systems that exist at this time step. Finally, repeat steps 1–8 as required: for fixed $\tau_{repeat}$, the total number of sills is specified a priori; for variable $\tau_{repeat}$, the calculation is continued until $\tau_{repeat}$ goes to infinity, thereby determining the total number of sills.

**Combining carbon emissions from the entire sill province.** We calculated carbon emissions flux and cumulative emitted carbon from the entire sill province by summing thermogenic and magma degassing emissions from many sill–vent systems (step 8 of the stochastic procedure). Say that at time $t$ after sill province initiation, $n$ sills have been intruded. The initiation times for these sills have already been determined sill-by-sill and are represented as $t_0(i)$, $i = 1, 2 \ldots n$. Time relative to intrusion of each sill is denoted as

$$\Delta t(i) = t - t_0(i) \, . \tag{31}$$

Emissions flux from the province is

$$Q_{prov}(t) = \sum_{i=1}^{n} \Big( q_{therm}(\Delta t(i)) + q_{magma}(\Delta t(i)) \Big) \tag{32}$$

and the cumulative mass is

$$M_{prov}(t) = \sum_{i=1}^{n} \Big( m_{therm}(\Delta t(i)) + m_{magma}(\Delta t(i)) \Big) \, . \tag{33}$$

The carbon isotopic composition of the emissions is

$$\delta^{13}C_{prov}(t) = \frac{1}{Q_{prov}(t)} \sum_{i=1}^{n} \Big( \delta^{13}C_{therm}(\Delta t(i)) q_{therm}(\Delta t(i)) + \delta^{13}C_{magma} q_{magma}(\Delta t(i)) \Big) \, . \tag{34}$$

**Isotopic composition of carbon emissions.** The isotopic composition of mantle-derived carbon is –7‰. The isotopic composition of thermogenic carbon depends on the composition and also the temperature of maturation of organic material in the host rock[52]. Both of these relationships are observed in the data compilation in Supplementary Fig. 4a. We used these data to define the relationships between $\delta^{13}C$ and vitrinite reflectance, $R_o$, a proxy for maturation temperature, for refractory (gas-prone) kerogen

$$\delta_{ref}^{13}C = 15.4 \log R_o - 41.3 \tag{35}$$

and labile (oil-prone) kerogen

$$\delta_{lab}^{13}C = 22.0 \log R_o - 32.0 \, . \tag{36}$$

We then used the coupled thermal-kinetic reaction model to determine the relationship between reaction temperature and vitrinite reflectance (Supplementary Fig. 4b). Combining the $\delta^{13}C(R_o)$ and $R_o(T)$ relationships gives the carbon isotopic composition as a function of reaction temperature $\delta^{13}C(T)$ (Supplementary Fig. 4c). The temperature of each reaction front can be tracked during numerical thermogenic reaction modelling, allowing us to estimate the carbon isotopic composition of thermogenic methane from both kerogen types over time for any sill (Supplementary Fig. 4d). These time-series show two phases. Between sill intrusion and full solidification, the carbon isotopic composition is relatively heavy and decreases little through time. At the time of full solidification, there is a kink in the composition curve, and thereafter the isotopic composition gets lighter over time. This behaviour arises because the temperature of the reaction front (where most of the methane flux is generated) decreases slowly until the sill solidifies fully and more rapidly thereafter until the background temperature is regained.

**Estimating sill intrusion repeat time from mantle plume flux.** There is as yet no effective method to determine the rate of LIP carbon emissions on a sub-thousand-year timeframe. One important difficulty is that it is impossible to measure the period $\tau_{repeat}$ between intrusion of successive sills directly using radiometric or biostratigraphic dating. Here we derive an alternative constraint on the sill intrusion repeat time $\tau_{repeat}$ based on the mantle convection process that generated the NAIP. If $n$ sills are intruded in time $t$, then $\tau_{repeat}$ and the corresponding frequency $f$ are

$$f = \frac{1}{\tau_{repeat}} = \frac{dn}{dt} \, . \tag{37}$$

To link $\tau_{repeat}$ to mantle plume flux, first expand this expression in terms of the surface area of the sill province $A_{LIP}(t)$ to give

$$f = \frac{dn}{dA_{LIP}} \frac{dA_{LIP}}{dt} \, . \tag{38}$$

We define the sill area-density (i.e. the number of sills per unit area) as

$$\rho = \frac{dn}{dA_{LIP}} \, . \tag{39}$$

If expansion of the NAIP sill province directly reflects sub-plate expansion of the hot mantle that generated the sill magma, then

$$Q = \frac{dA_{mantle}}{dt} = \frac{dA_{LIP}}{dt} \, , \tag{40}$$

where $Q$ is the mantle plume area flux, averaged vertically across the plume head of surface area $A_{mantle}$. In this case, the expression for intrusion frequency becomes

$$f = \frac{1}{\tau_{repeat}} = \rho Q \, . \tag{41}$$

More generally, expansion of the sill province is smaller than $Q$ because thick lithosphere inhibits decompression melting, so that $A_{LIP} < A_{mantle}$. This effect can

be incorporated by expanding in terms of plume head area to give

$$f = \frac{1}{\tau_{\text{repeat}}} = \rho \frac{dA_{\text{LIP}}}{dA_{\text{mantle}}} \frac{dA_{\text{mantle}}}{dt}. \tag{42}$$

A model of the plume head is required to link $dA_{\text{mantle}}/dt$ to $Q$. We used a simple kinematic model that has been previously applied to model Cenozoic development of the Iceland Plume[38], including estimating Paleocene–Eocene plume flux from dynamic support histories derived from sedimentary successions side of Scotland[37,38,43], and estimating Neogene plume flux from the V-Shaped Ridges of oceanic crust south of Iceland[34,35]. In this model, plume mantle flows radially outwards from a point source between two rigid parallel plates that represent the top and base of the asthenosphere. The original model for a radially symmetrical plume head is

$$A_{\text{mantle}} = \pi R^2 = 1.5Qt, \tag{43}$$

where $R$ is the radius of the plume head. We assume that the marker for the edge of the plume head is the location of maximum temperature anomaly, which is expected to be the location of peak melting, and to lie directly beneath the location of peak dynamic support of the overlying plate. The factor 1.5 occurs because this plume head marker travels at the maximum speed within the plume head channel, which is 1.5 times greater than the mean speed across the channel for a Poiseuille velocity profile[38]. If the plume head is elliptical, then the central term of Eq. 43 becomes $\pi \gamma R^2$, where $\gamma$ is the aspect ratio (i.e. the ratio of short and long radii) and $R$ is now the long radius of the ellipse. Differentiating gives

$$\frac{dA_{\text{mantle}}}{dt} = 1.5Q. \tag{44}$$

If we define a function

$$\Gamma = \frac{3}{2}\frac{dA_{\text{LIP}}}{dA_{\text{mantle}}} = \frac{3}{4\pi\gamma R}\frac{dA_{\text{LIP}}}{dR} \tag{45}$$

to relate the areal expansion of the plume head and the sill province, then the expression for the sill intrusion time period becomes

$$f = \frac{1}{\tau_{\text{repeat}}} = \rho(R)\Gamma(R)Q. \tag{46}$$

**Relating sill distribution and carbon emissions.** Although $\tau_{\text{repeat}}$, the time period between emplacement of successive sills, is a fundamental parameter that controls gas emissions flux $Q_{\text{prov}}$, $\tau_{\text{repeat}}$ is too small to measure directly using the geological record. It is therefore useful to derive an expression that relates $Q_{\text{prov}}$ to the spatial distribution of sills $\rho$, which can be measured directly. Results of stochastic modelling with constant sill recurrence time indicate an inverse relationship between $\tau_{\text{repeat}}$ and $Q_{\text{prov}}$ at long-time steady state (Eq. 1). The derivation in the previous section relates $\tau_{\text{repeat}}$ to $\rho$ (Eq. 46). Combining these two relationships eliminates $\tau_{\text{repeat}}$:

$$\rho = \frac{Q_{\text{prov}}}{1.65\,\Gamma\,Q}. \tag{47}$$

This expression can be used to investigate whether the $\rho$ observations are compatible with $Q_{\text{prov}}$ estimates obtained independently from sedimentary records of environmental change. Figure 7c shows the $\rho$ dataset overlain by a Gaussian curve that was used to calculate the NAIP emissions histories in Fig. 8. Figure 7c also shows an alternative $\rho$ curve, calculated using Eq. 47 with $Q_{\text{prov}}$ from a sedimentary record[5], the median value for $Q$, and $\Gamma$ from Fig. 7e. The two $\rho$ curves are quite different near the sill province centre, but they are both compatible with the data since $\rho$ measurements near the province centre are sparse. Supplementary Fig. 5 shows NAIP emissions flux histories calculated using both $\rho$ curves. Both a smooth rise to a lower peak flux and also a more irregular rise to a higher peak flux are probably compatible with available $\rho$ data at present, and it should be possible to distinguish between these two scenarios in future with more $\rho$ measurements near the province centre.

The NAIP $Q_{\text{prov}}$ model based on the alternative $\rho$ model is often lower than the sediment-based $Q_{\text{prov}}$ curve from which the alternative $\rho$ model was derived via Eq. 47 (Supplementary Fig. 5, purple versus blue). This discrepancy is expected for two reasons. First, Eq. 47 assumes long-time steady state via Eq. 1, so it should underestimate $\rho$ when $Q_{\text{prov}}$ increases rapidly. Secondly, Eq. 47 does not account for overlap between sill aureoles, so it should underestimate $\rho$ at later times. Nevertheless, Eq. 47 provides a useful new starting point for estimating the extent to which uncertainties in $\rho$ and also $\Gamma$ might explain differences between carbon source- and carbon sink-derived estimates of $Q_{\text{prov}}$.

**Determining sill locations.** Emission histories can be calculated without determining sill locations, and this procedure was used in the constant-$\tau_{\text{repeat}}$ calculations (Fig. 5). The constant-$\tau_{\text{repeat}}$ simulations represent maximum possible long-time emissions because they implicitly assume that each new sill intrudes host rock capable of generating methane. There are two reasons for assigning sill locations. First, the resulting maps provide a more intuitive description of our proposed link between mantle plume head and sill intrusion processes (Fig. 8g–j; Supplementary Movie 1). Secondly, sill locations can be used to ensure that when a sill intrudes host rock that has already been thermally matured by a previous sill, the emissions are not counted twice in the combined emissions histories.

To define a sill location, two of the random numbers that define the sill in step 2 of the stochastic modelling algorithm are used to choose the sill location radius and the sill location azimuth. Since melt generation is proportional to mantle

temperature, the sill location radius was chosen based on the location of the mantle thermal anomaly specified by the plume head model. Equation B14 of ref. [38] gives the temperature structure within the plume head channel as a function of plume head area, depth and time:

$$T(A_{\text{mantle}}, z, t) = \frac{1}{Q\delta\sqrt{2\pi}} \exp\left[\frac{-(A_{\text{mantle}}/Qf_z - t)^2}{2\delta^2}\right], \tag{48}$$

where $\delta$ is the standard deviation of the pulse of anomalously hot mantle and $f_z$ is a function that describes vertical variation in velocity across the plume head. If time is scaled so that $t = t'\delta$, plume head area is scaled so that $A_{\text{plume}} = A'Q\delta$ and $f_z$ is scaled by the thickness of the plume head channel, this equation becomes

$$T(A', z', t') = \frac{1}{(A_{\text{plume}}/A')\sqrt{2\pi}} \exp\left[\frac{-(A'/f'_z - t')^2}{2}\right], \tag{49}$$

where $f'_z = 1.5(1 - z'^2)$ for a Poiseuille flow profile[38]. This equation can be integrated numerically over depth to give the mean temperature anomaly as function of radius and time (Supplementary Fig. 6a). If this mean temperature function is integrated over radius and suitably scaled, the result can be interpreted as a cumulative probability distribution for the sill location radius (Supplementary Fig. 6b). In practice, we used area rather than radius for the horizontal coordinate and subtracted the position of the plume head marker (taken as the location of maximum temperature anomaly) in order to form a cumulative probability function for the sill location area anomaly $\delta A'$ for each sill (Supplementary Fig. 6d). This function was used as a look-up table to determine the sill location radius from the appropriate random number for each new sill. To create Figs. 8g–j and Supplementary Movie 1, we used $Q = 4$ km$^2$ yr$^{-1}$ (Fig. 6) and $\delta = 40$ kyr (ref. [38]).

After assigning a sill radius, an azimuth was selected from the ranges that lay within the deep basin chain (Fig. 6a), assuming a uniform distribution. Geographical coordinates were calculated from the (radius, azimuth) pair assuming the same elliptical plume head location and geometry used to determine mantle plume flux.

**Determining overlap between sill aureoles.** No new thermogenic methane is generated where the aureole of a newly intruding sill overlaps an existing sill or its aureole. This effect was modelled by modifying the $M_{\text{therm}}$ values of each sill. Each new sill was compared with existing sills in turn. The distance $d$ between the centres of each sill pair was determined from their radial coordinates. The sill footprints overlap partially if $d < (R_1 + R_2)$ and fully if $d < |R_1 - R_2|$, where the subscripts distinguish the two sills. For each partially overlapping sill pair, the area of the lens-shaped overlap is

$$\begin{aligned} A_O = {} & R_1^2 \cos^{-1}\left(\frac{d^2 + R_1^2 - R_2^2}{2dR_1}\right) + R_1^2 \cos^{-1}\left(\frac{d^2 + R_2^2 - R_1^2}{2dR_2}\right) \\ & - 0.5\sqrt{(R_1 + R_2 - d)(R_1 - R_2 + d)(R_2 - R_1 + d)(R_1 + R_2 + d)} \end{aligned} \tag{50}$$

The depths of the top and base of each sill are $Z \pm 0.5\,S$, and the depths of the top and base of the aureole were approximated by $Z \pm 0.5\,S\,(Y + 1)$. These depths were used to test for vertical overlap between aureoles analogous to the test for overlapping footprints. The overlapped aureole volume is $A_O Z_O$, where $Z_O$ is the vertical overlap. The overlapped volume as a proportion of the total aureole volume is

$$\varphi = \frac{A_O Z_O}{ASY}. \tag{51}$$

This fraction was used to update the total carbon mass parameter of each sill to $(1 - \varphi)M_{\text{therm}}$.

Two factors complicate this basic procedure. If a new sill overlaps multiple existing sills, it becomes awkward to determine $\varphi$ because the multiple overlapping volumes may overlap each other. It is not worth solving this problem in full bearing in mind the various simplifying assumptions already made, including circular sills, symmetrical aureoles and uniform initial geotherm. Instead, we note the probability that a point lying within a large volume $V$ is overlapped by at least one of $k$ small independent blobs each of volume $v$ is $1 - \left(1 - \frac{v}{V}\right)^k$. Hence an estimate of the factor to correct $M_{\text{therm}}$ for $k$ patches of aureole with fractional overlaps $\varphi_{1\ldots k}$ is

$$\varphi^* = 1 - \prod_{i=1}^{k}\left(1 - \varphi_i\right). \tag{52}$$

The second complicating issue is that each sill is divided into multiple annuli to account for radial variation in thickness. Therefore, the procedure above was carried out separately for each annulus, which requires extra steps to determine $A_O$ using the inner and outer radii of each intersecting pair of annuli.

## Data availability

Datasets generated by this study are available in two digital data repositories. Modelled emissions from individual sills (underlying Figs. 2, 3 and Supplementary Fig. 1) are available in GitHub repository smj75/sillburp with the identifier https://doi.org/10.5281/zenodo.3457994 (ref. [53]). Modelled emissions from the NAIP sill province (underlying Figs. 5 and 8 and Supplementary Fig. 5), sill dimension measurements (underlying Fig. 2), sill province area information and sill distribution measurements (underlying

Figs. 4, 6 and 7) are available in GitHub repository smj75/NAIPburp with the identifier https://doi.org/10.5281/zenodo.3454889 (ref. 54). The seismic reflection data used to measure sill dimensions and sill spatial distributions are owned by various third parties. The data owners and the means of accessing the data are listed in repository https://doi.org/10.5281/zenodo.3454889 (ref. 54). Some of these datasets are not publicly available and were used under license for this study, and requests to access these datasets must be addressed directly to the data owners.

## Code availability

Code generated by this study for modelling thermal kinetic reactions adjacent to individual sills is available in GitHub repository smj75/sillburp with the identifier https://doi.org/10.5281/zenodo.3457994 (ref. 53). Code generated by this study for stochastic simulation of combined LIP sill province emissions is available in GitHub repository smj75/LIPburp with the identifier https://doi.org/10.5281/zenodo.3458139 (ref. 55).

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

## Acknowledgements

We thank Spectrum Geo Ltd. for access to seismic reflection data to develop the sill dimension database. No external funding supported this project directly. M.H. was supported by UK Natural Environment Research Council (NERC) PhD studentship NE/1369185, T.D.J. was supported by NERC Standard Grant NE/P013112/1 and S.E.G. was supported by NERC Independent Research Fellowship NE/L011050 whilst working on this paper.

## Author contributions

S.M.J. conceived the research, developed the methodology, including new software, carried out the modelling and led the writing. M.H. assembled and interpreted the seismic database used to constrain sill dimensions. T.D.J. and S.E.G. advised on comparisons with PETM climate records and climate models, and assisted with writing.

## Competing interests

The authors declare no competing interests.
