## [Peer Review File · Nature Communications]

Reviewers' comments:

Reviewer #1 (Remarks to the Author):

This is an exciting and timely study that takes a novel approach to estimating the possible flux of carbon from sill intrusion associated with the North Atlantic Igneous Province, a mechanism that has been postulated as the driver of the Paleocene-Eocene Thermal Maximum. As a result, I think this study is appropriate for Nature Communications and will be of broad interest.

I want to state up front that I am not an expert in the dynamics of mantle convection or the mechanistic details relating sill formation to carbon flux. I find the approach the authors utilize for estimating emissions to be plausible and it seems to me that sufficient detail is provided between the Methods and Supplemental text to provide justification for assumptions made during modeling (or at least it would be easy for an expert reader to understand what assumptions were made and on the basis of what data). Instead, my suggestions for the manuscript are mainly to improve its accessibility for the community of PETM and paleoclimate researchers more broadly. My impression is that the main text is quite short relative to the length restrictions of Nature Communications, and there is information in the online-only Methods and Supplemental text that should be moved to the main text. This is particularly the case with the section 'Modeling NAIP carbon emissions flux' on pg. 7. I think that this section is the most novel and exciting part of the paper, leading to mechanistic estimates of the total mass and timescale of carbon emissions from NAIP sill intrusion, but the details needed to understand these results are not outlined here. The parameterizations for $m(t)$ (the cumulative mass of carbon generated as a function of time since sill intrusion) and $q_{\text{therm}}(t)$ (the associated flux) (Section S2) seem crucial, and as these are empirical parameterizations that do not seem to be broadly used (or published elsewhere), I think they belong in the main text. Even the itemized list of steps in the stochastic modeling procedure provided in the Methods did not provide these parameterizations, and I did not know exactly what q and m in Step 8 referred to until I got through the supplement. Then, the terms that suggest how these masses and fluxes for individual sills are summed for the entire sill province (Section S5) are also particularly important. In fact, the first paragraph in Section S5 would have been very helpful in the section on modeling emissions fluxes in the main text. I found it difficult to connect the methodology described in the modeling emissions flux section to the methodology in the previous section 'Linking mantle plume flux and sill intrusion frequency.' It was not clear to me how the basic expression for the sill intrusion recurrence period given in that section could be used to generate a timeline of emissions fluxes. Then, once the authors indicated that the mantle plume head model could also specify how the repeat time varied with time in the following section, I was left wondering about the form of those equations and the basic steps in this modeling. I think these steps (provided in more detail in methods) could be better summarized in a short paragraph in the main text.

Another detail that I think is of particular interest to PETM researchers that has been left in the supplement is the isotopic composition of the carbon emissions associated with sill intrusion. One of the reasons that the community has remained skeptical about volcanic sources for the PETM is the assumption of too heavy isotopic compositions given a canonical mantle value of around -6 per mil. I think there is a general understanding that thermogenic methane should have a much more depleted isotopic signature than typical mantle, but I didn't find the word 'thermogenic' used at all in the main text (despite the fact that it is mentioned consistently in the Methods and Supplement). It is important that the modeled carbon isotopic composition of the flux is consistent with requirements for generating the CIE, since the authors compare their emissions estimates to inverse and forward modeling of PETM carbon emissions that are both inextricably tied to the CIE. Points like '95% of the carbon emissions from the sill province come from thermogenic methane, with the remainder from mantle-derived carbon dioxide' really belong in the main text. The final goal of the paper, indicated in the abstract, to 'open up the possibility of reconstructing the magnitude of climate-carbon cycle feedbacks' will hinge on constraints on both the mass and the isotopic composition of the carbon emissions necessary to explain PETM records. The results presented in the supplement for this study (Figure S7) seem to indicate that the sill-carbon emissions also match the required isotopic composition (at least from the Gutjahr study; the Frieling study assumed a -50 per mil initial input composition). I think that the disagreement within the community about the required isotopic composition of the initial emissions provides a

justification if the authors do not want to indicate the difference between their calculated emissions and those diagnosed or suggested in other studies with the assumption that the residual is due to carbon cycle feedbacks, but I think it is a missed opportunity to keep these results for the isotopic composition of their calculated flux in the Supplement.

Finally, there are a few details on figures that I think require clarification. In figure 1f, what is the difference between the light grey and dark grey shaded regions around the thin and thick black lines? What is the difference between the inverted and upright triangles? In the figure 2 caption, why are the grey lines in panel (a) not defined? In figure 3, the text is very small and it is difficult to read in panels d through f. Also in figure 3a-c, what are the red dashed and dotted lines (I'm assuming these are confidence intervals of some sort on the 100 simulated emissions histories)?

Reviewer #2 (Remarks to the Author):

Review of 'Large Igneous Province greenhouse gas flux can initiate Paleocene-Eocene Thermal Maximum' by Jones, Hoggett, Greene and Dunkley Jones.

Jones et al. describe a new approach to calculate the mass and emission flux of carbon during emplacement and cooling of sills formed in the North Atlantic Igneous Province, during the Paleocene-Eocene Thermal Maximum (PETM). Their simulations predict sill emplacement and associated thermal maturation of host rocks to predict carbon emissions. They show that the emplacement of sills and their impact on the organic maturation of host rocks were sufficient to generate and probably emit enough carbon to initiate PETM climate change. The topics tackled in this paper are interest to a wide audience including climate scientists, geologists and geochemists.

They point out that to obtain direct measurements of the timing and amount of igneous material (i.e. sills) emplaced during the PETM is challenging because seismological, borehole and radiometric dating do not have the required resolution. Instead they show how models of mantle plume evolution combined with empirical scaling of sill emplacement and thermo-kinetic modeling of host rock maturation can simulate carbon production in the thermal aureoles of sills. They tested a suite of plausible and parsimonious models to generate a convincing thesis about a poorly understood contributor to global climate change. They discuss caveats regarding the relationship between generated and emitted carbon in the supplementary text. The uncertainties in model parameters (e.g. sill size, plume head shape, maturation) are discussed and propagated through their models. The paper is well written, it has a good structure and the supplementary information, including the movie, go a long way to helping the reader understand the methodologies. As such, I think this thought provoking and careful study, should be published.

My comments and criticisms do not focus on the methodologies presented, which I think are appropriate and well tested. Instead, they center on tweaking the description and presentation of data, methodologies and results, which could be modified or expanded in places to help the reader understand and reproduce results more easily.

Comments on methodology

Having said that I have two minor questions about the methodology. First, the authors mention that the knowing the location of the emplaced sills is unimportant for calculating carbon emission histories (Supplementary text S7). However, as they acknowledge, if a sill is emplaced close to (within the thermal aureole) of an existing sill that would change (reduce) the amount of carbon emitted by the second sill. Although they state that the sill locations could be used to avoid 'double counting' of carbon emission it is not clear to me from section S7 that they actually did so in this study? Secondly, it seems to me the most challenging assumption in this study is that carbon escapes the solid Earth (i.e. gets through overburden) once it has been produced. The authors do discuss the issue at some length in the supplementary text (S3) but it would be helpful if they could also give an indication of the rates of escape expected in different scenarios (e.g. tight/fractured overburden) and discuss if/how those scenarios pertain to the NAIP. In other words, can the authors place quantitative constraints on the expected rate of methane escape and are they generally expected to be fast enough that they have no bearing on the results of this study?

Comments on presentation

First, the authors could more clearly show the data (or a subset) used to parameterize the statistical models of sill distribution. For example, they could show (at least) one of the seismic images discussed that contains evidence for a sill in the NAIP and some information about how they depth converted that data to obtain sill thicknesses. I think that including such a figure, and perhaps an accompanying schematic, would allow the reader to understand the data used to parameterize the sill models and associated maturation models more easily. If space permits I think this figure, with perhaps a map of the study region, would give a much more accessible introduction to the paper than the current Figure 1.

Second, in the main manuscript, the authors state that 'the maximum carbon flux generated by one sill is considerably less, at 0.0005 to 0.005 PgC yr⁻¹' and refer the reader to Figure 1a, which refers the reader to the Methods section. I find it hard to understand what part of the Methods section I am supposed to use to determine the maximum carbon flux of a single sill. I think that it would help the reader to follow this part of the main manuscript more easily if it included a brief description of how maximum annual carbon flux was estimated (e.g. 'integration of flux calculated from coupled thermo-kinetic reaction calculations show that...'). It seems crucial to me that the reader believes these numbers are reasonable whilst reading the body text because of what comes next.

Third, in my opinion, the final section of the manuscript could be more focused on explaining the authors' novel approach and results, and their broader implications (e.g. that mantle convection can instigate rapid and extreme climate change). I think that in its current form it is somewhat difficult to pick out what the authors consider the most important and robust findings.

Fourth, I find the order of the figures confusing. As I said above, I think a map and images showing seismic and well data (with sills) and a schematic showing the setup of the sill-maturation model would help to clarify the rationale and methodologies. I think that the current Figure 2, which describes the sill observations, should come before the modeling of the sill emplacement and related carbon generation/emission (current Figure 1). I think that it would help to reader to understand the universality of the model if the sills from the NAIP and elsewhere were given different symbols/colors in Figure 3c. I also think that Figure 4, which shows the setup of the solid Earth (mantle convection and sill emplacement) models, should come before Figure 3, which contains the main results. The maps shown in Figure 3 are very small and the pie chart/legend annotations are barely legible on my version, please enlarge.

Fifth, I find the start of the Methods section confusing. What is the enumerated list supposed to be telling the reader, is it a précis of the methodology for the entire study? It is very terse and in places is difficult to follow or make use of. For example, the statement about 'drawing 8 random numbers' referred to in item 2 is too cryptic for me; what do the 8 numbers pertain to? In item 4, it is not clear to me how I could 'define the weight fraction that converts methane... and the kerogen composition parameter K using the cumulative probability distribution in Fig. 2'. I think the K parameter, and beta in item 3, need to be defined more thoroughly here or the reader be referred to the supplementary information earlier. I think it would help if this list referred the reader to the relevant detailed descriptions of the methodologies given in the supplementary text more.

Sixth, the methodologies are described in the supplementary text in detail and with care, but I think that they could be referred to in the Methods section of the paper earlier. I've a few suggested tweaks/corrections to the supplementary text. (1) In the third paragraph, correct 'may to overestimate'. (2) I suggest giving the conductive cooling PDE in full, partly because the conductive cooling model is central to the thermal evolution calculations, but also because it contains complicating terms (i.e. the latent heat of crystallization). I think its inclusion would help the reader to understand Figure S4 more easily. (3) What are the horizontal and vertical lines in Figure S2b showing? (4) I think 'blue circles' should be 'blue squares' in the caption to Figure S3b? (5) I cannot see the green curves referred to in the caption to Figure S7.

Reviewer #3 (Remarks to the Author):

The paper presents the results of modeling of carbon emissions caused by the North Atlantic Igneous Province causing a significant warming event at the Paleocene-Eocene boundary. The model links the intrusion of sills in Mesozoic sediments by basaltic magma generated by the North Atlantic mantle plume, using information on the speed and relative movement of the plume to constrain the speed of generation of the sill complexes. This information is then used to model the production of carbon gas from the intruded sediments (and the sills?). The carbon excess injected in the atmosphere has been linked to the extreme warming.

The study is interesting as it quantifies and discusses many of the geological variables controlling the final carbon released and the relative timing and tempo of its release. As such it would make a good addition to the growing literature dedicated to this event and also very relevant with respect to the present global warming.

My problem with the paper is that the main body of the paper tends to be very diffuse and not very quantitative. I will discuss some of the examples below. The more substantial part of the paper is the Method section and the Supplementary parts.

The title is much generalized. It indicates the general topic of the paper, but it is not very specific on the substance of the study, specifically the attempt to quantify amount and speed of carbon gas generation to sill complexes and to the mantle plume.

The abstract: I had to stop and reflect on the meaning of several sentences. Some examples:

Line 2: 'Initiation of the North Atlantic Igneous Province (NAIP) spanned the PETM...'. Why initiation? It implies that it was followed by a main phase and then by a termination? How is the initiation defined with respect to the period of emplacement of the sills? How long did the PETM last?

Line 5: '... the multi-millennial timeframe of PETM onset'. As above, what is the PETM onset? Was it followed by a main period and then a termination phase?

Line 11: 'the onset period'. Again, what is that?

Line 17: 'magma associated carbon source'. Suggests that the carbon is released by the magma, in reality most of the carbon is modelled as coming from the intruded sediment. Nobody would ever guess this by reading the abstract. Thus, I find the abstract mainly very suggestive but very little informative.

Main text, first sentence: 'A temporal association between Large Igneous Provinces (LIPs) and perturbations to global climate and the carbon cycle occurs throughout Mesozoic time...' The statement could be interpreted as this having been a continuous process, with some more specific stronger episodes.

page 3, line 3: 'a rapid period' time is not rapid or slow. You mean short period?

Line 4: 'Although the North Atlantic Igneous Province (NAIP) LIP and the PETM are closely coincident in time, the rate of NAIP carbon emissions has not yet been reconciled with the rapid onset of PETM climate change.' Very generalized statement: please state what 'the rate of NAIP carbon emissions' is, and quantify what the 'rapid onset of PETM climate change' actually was (increase of x °C over y years).

End of second paragraph on page 3: 'However, all these theories struggle because their timeframes have not been shown to match the timeframe of the PETM onset. Consequently, the NAIP has typically been relegated to a driver of longer term background warming, which perhaps triggered more rapid release of carbon from other temperature-sensitive near-surface reservoirs around the globe'. It seems to me that it overgeneralizes. Some of the quoted papers seem to

draw quite a direct link between PETM and magmatism. Again a more specific and less generalized presentation should mention the actual arguments and mechanisms proposed in the literature, so the reader would know exactly what the authors refer to.

3rd line, 3rd para, page 3: '...these sills are the most likely source of the large total mass of carbon...'. The model assumes that the bulk of the C was released from the sediments; just 5% came from the magma. This sentence suggests the opposite.

page 4, line 6: 'cooling time period for the sill, of order 100s to 1000s years...'

page 4, 2nd para, line 2: Please define 'the PETM onset period'

page 6, 2nd para, line 2: 'The mantle plume flux parameter is a fundamental measure of how fast temperature pulses move round a mantle convection cell'. The variable has a dimension of [surface over time] and seems to me to indicate the progression of the plume with respect to the overlying plate, but a more concrete explanation would certainly be useful if the paper has to be read also by non-specialists.

These examples reflect the style of the paper. Very often using generalizations that lack more specific references and arguments, and are often very qualitative, lacking quantitative back-up. It is almost like the authors want to keep things as diffuse as possible. Perhaps adding a figure that illustrates the relationships between plume and sills and sediments and rift, pointing out the various parameters used in the model, would help.

Fig. 2: I suggest enlarging a-c, where now the details and labeling of far too small. I would then remove d-f, the symbols and letter are so small that they are next impossible to decipher. And the movie shows the distribution very well.

In chapter 4 of the supplementary material there is a discussion of the C in the magma. It is interesting, but I was also left in some confusion. Since the total C released is important, it is not evident to me why lavas and lower crustal bodies are not included. The reasoning in that paragraph seems to contradict itself a bit. But such information with estimated quantities and length of processes should clearly be given in the main text.

F. Corfu 2-5-2019

Summary of Changes

Title changed slightly to avoid active verb.

Abstract shortened and references removed.

Introduction lengthened to c. 1000 words. The first 3 introductory paragraphs are retained from previous version. The 4th paragraph condenses material from the original main text section “Sill intrusion frequency required to explain PETM”. The final paragraph is new and summarises the content and main conclusion, as per NComms requirements.

Results contains following subsections.

Thermogenic carbon emissions procedure (results parts consolidated from original Methods section “Thermogenic carbon emissions parameterization” and supplementary section S1).

Magmatic carbon emissions procedure (results part of original supplementary section S4).

Sill and host rock observations database (results part of original Methods “Sill and hoist rock observations”).

Combined LIP sill province carbon emissions (results part of original supplementary section S5).

Derivation of link between trepeat and mantle plume flux (results part of original Methods “Relating sill intrusion repeat time to mantle plume flux”).

Mantle plume flux observations (results part of original Methods “Mantle plume flux”).

NAIP sill province geography (results part of original Methods section “Sill intrusion density”).

Time-dependent model of NAIP carbon emissions flux (previously part of main text section “Modelling NAIP Carbon Emissions Flux”).

Discussion has no sub-sections, but is organised as follows.

1. Paragraph to explain rapid onset of high emissions flux (material from original supplementary section S4).
2. Paragraph to explain post-peak decline in emissions flux (material from original main text section “Modelling NAIP Carbon Emissions Flux” augmented as suggested by reviewer 2).
3. Paragraph to discuss proportion of Thermogenic carbon emissions (material from original supplementary section S5).
4. Paragraph to discuss carbon isotopic composition of emissions (material from original supplementary section S4).
5. Paragraph to discuss relationship between carbon from the sill province and additional carbon generated by the magma-rich passive margins (material from original supplementary section S4).
6. Paragraph to discuss how much of the generated gas is emitted (material from original supplementary section S3).
7. Concluding paragraph (modified from original concluding paragraph).

Methods has the following subsections.

Kinetic reaction modelling of thermogenic emissions (part of original supplementary section S1).

Parameterisation of thermogenic emissions results (original supplementary section S2).

Parameterisation of carbon emissions from magma degassing (part of original supplementary section S4);

Sill and host rock observations database (part of original Methods “Sill and host rock observations”).

Stochastic carbon emissions modelling procedure (originally Methods “Stochastic modelling procedure”).

Combining carbon emissions from the entire sill province (part of original supplementary section S5).

Isotopic composition of carbon emissions (part of original supplementary section S6).

Estimating sill intrusion repeat time from mantle plume flux (original Methods “Relating sill intrusion repeat time to mantle plume flux”).

Determining sill locations (original supplementary section S7).

Determine overlap between sill aureoles (new in response to reviewer 2).

Data and Code Availability: new section.

Figure 1: new introductory figure to illustrate concept.

Figure 2: previous Fig. S4.

Figure 3: previous Fig. S5.

Figure 4: previous Fig. 2.

Figure 5: previous Fig. 1.

Figure 6: previous Figs S1 and S3 combined.

Figure 7: previous Fig. 4 augmented.

Figure 8: previous Fig. 3 and Fig S7 combined.

Table 1: previous Table S1.

Table 2: previous Table S2.

Supplementary Movie 1: as before

Supplementary Figure 1: previous Fig. S6.

Supplementary Figure 2: new in response to reviewer.

Supplementary Figure 3: previous Fig. S2.

Supplementary Figure 4: previous Fig. S8.

Supplementary Figure 5: previous Fig. S9.

Supplementary Table 1: new notation table.

Supplementary Table 2: previous Table S3.

Reviewer 1: point-by-point responses.

This is an exciting and timely study that takes a novel approach to estimating the possible flux of carbon from sill intrusion associated with the North Atlantic Igneous Province, a mechanism that has been postulated as the driver of the Paleocene-Eocene Thermal Maximum. As a result, I think this study is appropriate for Nature Communications and will be of broad interest

I want to state up front that I am not an expert in the dynamics of mantle convection or the mechanistic details relating sill formation to carbon flux. I find the approach the authors utilize for estimating emissions to be plausible and it seems to me that sufficient detail is provided between the Methods and Supplemental text to provide justification for assumptions made during modeling (or at least it would be easy for an expert reader to understand what assumptions were made and on the basis of what data). Instead, my suggestions for the manuscript are mainly to improve its accessibility for the community of PETM and paleoclimate researchers more broadly.

We are grateful to the reviewer's suggestions to improve accessibility for palaeoclimate researchers. We have implemented all of them.

"My impression is that the main text is quite short relative to the length restrictions of Nature Communications, and there is information in the online-only Methods and Supplemental text that should be moved to the main text.

Main text lengthened to c. 4900 words and contains all of the previous supplementary text. Supplementary section of the present draft contains no text, only figures.

"This is particularly the case with the section 'Modeling NAIP carbon emissions flux' on pg. 7. I think that this section is the most novel and exciting part of the paper, leading to mechanistic estimates of the total mass and timescale of carbon emissions from NAIP sill intrusion, but the details needed to understand these results are not outlined here. The parameterizations for $m(t)$ (the cumulative mass of carbon generated as a function of time since sill intrusion) and $q_{\text{therm}}(t)$ (the associated flux) (Section S2) seem crucial, and as these are empirical parameterizations that do not seem to be broadly used (or published elsewhere), I think they belong in the main text.

The reviewer notes that this is a new parameterisation, not previously published. We make this point more clearly in the introduction (L93). The motivation for the new parameterisation, the way it works (in English, without equations) and the main results are now described in the first sub-section of the main-text Results, headed "Thermogenic carbon emissions flux parameterisation procedure" (L111). The details and equations are in Methods sub-sections "Kinetic reaction modelling of thermogenic emissions" (L409) and "Parameterisation of thermogenic emissions results" (L504).

"Even the itemized list of steps in the stochastic modeling procedure provided in the Methods did not provide these parameterizations, and I did not know exactly what q and m in Step 8 referred to until I got through the supplement. Then, the terms that suggest how these masses and fluxes for individual sills are summed for the entire sill

province (Section S5) are also particularly important. In fact, the first paragraph in Section S5 would have been very helpful in the section on modeling emissions fluxes in the main text.

The stochastic modelling procedure is now introduced toward the end of the Introduction (L96). The modelling procedure is broken up into 3 stages (thermogenic emissions parameterisation, sill dimension database, combining multiple emissions) that correspond to sub-sections in the Results (L111, L162, L171). Terms such as q , m are now defined in the first of these sub-sections, where they first appear (L136). The itemised list of pseudo-code for stochastic modelling of emissions from multiple sill-vent complexes (which headed the Methods section of the previous draft) is now in the new Methods sub-section "Stochastic carbon emissions modelling procedure" (L630). The summation equations previously in §5 of the Supplementary Material are now in Methods sub-section "Combining carbon emissions from their entire sill province" (L659). Introductory sentences from starting paragraph in §S5 of the previous version are re-located to the Results sub-section "Combined LIP sill province carbon emissions" (L207).

"I found it difficult to connect the methodology described in the modeling emissions flux section to the methodology in the previous section 'Linking mantle plume flux and sill intrusion frequency.' It was not clear to me how the basic expression for the sill intrusion recurrence period given in that section could be used to generate a timeline of emissions fluxes. Then, once the authors indicated that the mantle plume head model could also specify how the repeat time varied with time in the following section, I was left wondering about the form of those equations and the basic steps in this modeling. I think these steps (provided in more detail in methods) could be better summarized in a short paragraph in the main text.

The order that the methodology is presented has been revised. The final paragraph of the Introduction (L87) now explains that the problem is broken down into 2 stages: (1) determining the sill recurrence period that would be required to give emissions sufficient to initiate the PETM (which turns out to be 2–6 years based on stochastic modelling); and (2) demonstrating that the NAIP sills actually did intrude at periods of 2–6 years. This paragraph also explains that these two stages are further broken down into steps that correspond directly to sub-sections of the Results. Derivation of the basic equations for sill recurrence period in terms of mantle plume flux make up one of these steps: it is introduced in L100, the main points and equations are given in Results sub-section "Derivation of link between τ_{repeat} and mantle plume flux" (L221), and the full derivation is provided in Methods sub-section "Estimating sill intrusion repeat time from mantle plume flux" (L693).

"Another detail that I think is of particular interest to PETM researchers that has been left in the supplement is the isotopic composition of the carbon emissions associated with sill intrusion. One of the reasons that the community has remained skeptical about volcanic sources for the PETM is the assumption of too heavy isotopic compositions given a canonical mantle value of around -6 per mil. I think there is a general understanding that thermogenic methane should have a much more depleted isotopic signature than typical mantle, but I didn't find the word 'thermogenic' used at all in the main text (despite the fact that it is mentioned consistently in the Methods and Supplement). It is important that the modeled carbon

isotopic composition of the flux is consistent with requirements for generating the CIE, since the authors compare their emissions estimates to inverse and forward modeling of PETM carbon emissions that are both inextricably tied to the CIE. Points like '95% of the carbon emissions from the sill province come from thermogenic methane, with the remainder from mantle-derived carbon dioxide' really belong in the main text. The final goal of the paper, indicated in the abstract, to 'open up the possibility of reconstructing the magnitude of climate-carbon cycle feedbacks' will hinge on constraints on both the mass and the isotopic composition of the carbon emissions necessary to explain PETM records. The results presented in the supplement for this study (Figure S7) seem to indicate that the sill-carbon emissions also match the required isotopic composition (at least from the Gutjahr study; the Frieling study assumed a -50 per mil initial input composition). I think that the disagreement within the community about the required isotopic composition of the initial emissions provides a justification if the authors do not want to indicate the difference between their calculated emissions and those diagnosed or suggested in other studies with the assumption that the residual is due to carbon cycle feedbacks, but I think it is a missed opportunity to keep these results for the isotopic composition of their calculated flux in the Supplement.

The word "thermogenic" is now used in the Title and throughout the main text, including the Introduction. The reviewer notes that there is disagreement within the community about the required isotopic composition of the initial emissions. Our main concern in this paper is to determine geologically plausible emissions fluxes, and we note that surface ocean pH records provide a tighter constraint on emissions flux than carbon isotopic compositions do (L348). Therefore, we do not discuss the isotopic composition in the main-text Results section. Instead, we have devoted a paragraph of the main text Discussion to this topic (L348). The result that 95% of the carbon emissions from the sill province come from thermogenic methane, with the remainder from mantle-derived carbon dioxide is also explored in a Discussion paragraph (L340). We retain the Discussion paragraph on the difference between the calculated emissions and those diagnosed or suggested in other studies, and the possibility that the residual is a measure of carbon cycle feedbacks (L392). The details of our new thermogenic carbon isotope parameterisation are now in the Methods sub-section "Isotopic composition of emissions" (L672). The summation equation for isotopic composition of emissions (missing from the original draft) is now included in Methods sub-section "Combining carbon emissions from the entire sill province" (L659).

"Finally, there are a few details on figures that I think require clarification. In figure 1f, what is the difference between the light grey and dark grey shaded regions around the thin and thick black lines? What is the difference between the inverted and upright triangles?"

This is Fig. 5 of the revised draft. Caption now states that the light and dark grey shaded regions are confidence intervals (± 1 s.d. for the 100-run ensembles). Caption now references the inverted and upright triangles separately.

"In the figure 2 caption, why are the grey lines in panel (a) not defined?"

This is Fig. 4 of the revised draft. These details were already summarised on the figure panel itself and they are now clarified in the caption.

"In figure 3, the text is very small and it is difficult to read in panels d through f. Also in figure 3a-c, what are the red dashed and dotted lines (I'm assuming these are confidence intervals of some sort on the 100 simulated emissions histories)?"

This is Fig. 8 of the revised draft. Text now enlarged. Red dashed and dotted lines now explained in caption.

Reviewer 2: point-by-point responses

Jones et al. describe a new approach to calculate the mass and emission flux of carbon during emplacement and cooling of sills formed in the North Atlantic Igneous Province, during the Paleocene-Eocene Thermal Maximum (PETM). Their simulations predict sill emplacement and associated thermal maturation of host rocks to predict carbon emissions. They show that the emplacement of sills and their impact on the organic maturation of host rocks were sufficient to generate and probably emit enough carbon to initiate PETM climate change. The topics tackled in this paper are interest to a wide audience including climate scientists, geologists and geochemists.

They point out that to obtain direct measurements of the timing and amount of igneous material (i.e. sills) emplaced during the PETM is challenging because seismological, borehole and radiometric dating do not have the required resolution. Instead they show how models of mantle plume evolution combined with empirical scaling of sill emplacement and thermo-kinetic modeling of host rock maturation can simulate carbon production in the thermal aureoles of sills. They tested a suite of plausible and parsimonious models to generate a convincing thesis about a poorly understood contributor to global climate change. They discuss caveats regarding the relationship between generated and emitted carbon in the supplementary text. The uncertainties in model parameters (e.g. sill size, plume head shape, maturation) are discussed and propagated through their models.

My comments and criticisms do not focus on the methodologies presented, which I think are appropriate and well tested. Instead, they center on tweaking the description and presentation of data, methodologies and results, which could be modified or expanded in places to help the reader understand and reproduce results more easily.

Similar comments are made by Reviewer #1 and the editor. We have adopted all suggestions on tweaking description and presentation of data.

Comments on methodology

Having said that I have two minor questions about the methodology. First, the authors mention that the knowing the location of the emplaced sills is unimportant for calculating carbon emission histories (Supplementary text S7). However, as they acknowledge, if a sill is emplaced close to (within the thermal aureole) of an existing sill that would change (reduce) the amount of carbon emitted by the second sill. Although they state that the sill locations could be used to avoid 'double counting' of carbon emission it is not clear to me from section S7 that they actually did so in this study?

We agree with the reviewer. The original calculations did allow potential for double counting. In the present version, the full NAIP simulations (variable repeat time) have been adjusted to avoid double counting and all figures and text updated accordingly. The original calculations are retained for the constant repeat time calculations. The

method of avoiding double counting is explained in the Methods subsection "Determining overlap between sill aureoles". The effect is to reduce emissions in at times > 10 kyr; this is illustrated in new Fig. 8f and discussed in a new paragraph of the discussion.

Secondly, it seems to me the most challenging assumption is this study is that carbon escapes the solid Earth (i.e. gets through overburden) once it has been produced. The authors do discuss the issue at some length in the supplementary text (S3) but it would be helpful if they could also give an indication of the rates of escape expected in different scenarios (e.g. tight/fractured overburden) and discuss if/how those scenarios pertain to the NAIP. In other words, can the authors place quantitative constraints on the expected rate of methane escape and are they generally expected to be fast enough that they have no bearing on the results of this study?

Again, we agree with the reviewer. This is probably the biggest uncertainty in our calculations. It is barely addressed in the literature, and has yet to be satisfactorily addressed in our opinion. All of the material previously in §S3 "Relationship between Generated and Emitted Methane" is re-located to its own paragraph in the Discussion of the present manuscript (L376). We state the observations that support the common assumption that methane escapes efficiently (L379). We explain in this section the steps that will be required to quantify the rates of escape in future (L385), and we state that the tools to carry out these steps have not yet been developed. (We do plan to develop them but this is a full project in itself and cannot be done within the 3-month turnaround time for this resubmission).

Comments on presentation

First, the authors could more clearly show the data (or a subset) used to parameterize the statistical models of sill distribution. For example, they could show (at least) one of the seismic images discussed that contains evidence for a sill in the NAIP and some information about how they depth converted that data to obtain sill thicknesses. I think that including such a figure, and perhaps an accompanying schematic, would allow the reader to understand the data used to parameterize the sill models and associated maturation models more easily. If space permits I think this figure, with perhaps a map of the study region, would give a much more accessible introduction to the paper than the current Figure 1.

A new schematic is provided as Fig. 1, which includes a perspective map view of the NAIP sill province study region, and shows how the sill province relates to the mantle processes that drive the melting. We provide a new figure to show an example of sills imaged as imaged on seismic data, explain how they are measured, and how the data is depth converted (Supplementary Fig. 2). We have added a panel to the sill and host-rock data figure (now Fig. 4; was Fig. 2) to clarify the various sill dimensions whose probability distributions are shown.

Second, in the main manuscript, the authors state that 'the maximum carbon flux generated by one sill is considerably less, at 0.0005 to 0.005 PgC yr⁻¹' and refer the reader to Figure 1a, which refers the reader to the Methods section. I find it hard to understand what part of the Methods section I am supposed to use to determine the maximum carbon flux of a single sill. I think that it would help the

reader to follow this part of the main manuscript more easily if it included a brief description of how maximum annual carbon flux was estimated (e.g. 'integration of flux calculated from coupled thermo-kinetic reaction calculations show that...'). It seems crucial to me that the reader believes these numbers are reasonable whilst reading the body text because of what comes next.

The equivalent sentence starts on L66 of the main text in the present version. I have altered it to obtain a generic carbon flux estimate from published work, to avoid the original problem of looking forward to as yet unexplained parts of the present manuscript. Fig. 1a in the previous draft is Fig. 5a in the current manuscript. This panel is now described in Results sub-section "Combined LIP sill province carbon emissions" (L207). This material now comes after the sub-sections of the results that described the emissions parameterisation and the sill observations, so it should be clearer how the numbers were produced.

Third, in my opinion, the final section of the manuscript could be more focused on explaining the authors' novel approach and results, and their broader implications (e.g. that mantle convection can instigate rapid and extreme climate change). I think that in its current form it is somewhat difficult to pick out what the authors consider the most important and robust findings.

The Discussion section has been re-organised and lengthened to take advantage of Nat. Comm.'s format. It tackles the following subjects, which were all mentioned in the methods and/or supplementary sections of the previous draft. (1) Explanation of how the NAIP sill province can supply an rapid initial increase in carbon emissions flux, sufficient to explain PETM initiation, in terms of mantle processes (L321). (2) Explanation of post-peak decline in emissions flux in terms of mantle processes (L331). (3) Discussion of proportion of thermogenic to mantle-derived carbon emissions (L341), comparison of predicted carbon isotopic composition of emissions with that inferred from climate records (L348), and discussion of relationship between carbon from the sill province and additional carbon generated by the magma-rich passive margins (L362). (4) Discussion of how much of the generated gas is emitted, and a statement that we would like to see more work on this topic. (5) Concluding paragraph exploring implications of new work for measuring carbon cycle feedbacks in future.

Fourth, I find the order of the figures confusing. As I said above, I think a map and images showing seismic and well data (with sills) and a schematic showing the setup of the sill-maturation model would help to clarify the rationale and methodologies. I think that the current Figure 2, which describes the sill observations, should come before the modeling of the sill emplacement and related carbon generation/emission (current Figure 1). I think that it would help to reader to understand the universality of the model if the sills from the NAIP and elsewhere were given different symbols/colors in Figure 3c. I also think that Figure 4, which shows the setup of the solid Earth (mantle convection and sill emplacement) models, should come before Figure 3, which contains the main results. The maps shown in Figure 3 are very small and the pie chart/legend annotations are barely legible on my version, please enlarge.

A new schematic is provided as Fig. 1, which includes a perspective map view of the NAIP sill province, and shows how the sill province relates to the mantle processes that drive the melting. A schematic representation of the magma transport system and sills.

We now include 2D and 3D seismic images of sills in the sill and host-rock data figure (now Fig. 4; was Fig. 2).

Fig. 2 in the previous draft is Fig. 4 in the present draft. It comes before Fig. 1 in the previous draft, which is Fig. 5 in the present draft.

Fig. 4 in the previous draft is Fig. 7 in the present draft. It comes before Fig. 3 in the previous draft, which is Fig. 8 in the present draft.

Text on Fig. 8 (previously Fig. 3) now enlarged.

Fifth, I find the start of the Methods section confusing. What is the enumerated list supposed to be telling the reader, is it a précis of the methodology for the entire study? It is very terse and in places is difficult to follow or make use of. For example, the statement about 'drawing 8 random numbers' referred to in item 2 is too cryptic for me; what do the 8 numbers pertain to? In item 4, it is not clear to me how I could 'define the weight fraction that converts methane... and the kerogen composition parameter K using the cumulative probability distribution in Fig. 2'. I think the K parameter, and beta in item 3, need to be defined more thoroughly here or the reader be referred to the supplementary information earlier. I think it would help if this list referred the reader to the relevant detailed descriptions of the methodologies given in the supplementary text more.

This material is now summarised in Results sub-section "Combined LIP sill province carbon emissions" (L207). The pseudo-code list is retained in Methods sub-section "Stochastic carbon emissions modelling procedure" (L630); at this point the various symbols have previously been defined where they first occur, in the Results and earlier Methods. A notation table to define all variables used in the equations is provided as Supplementary Table 1.

Sixth, the methodologies are described in the supplementary text in detail and with care, but I think that they could be referred to in the Methods section of the paper earlier.

The order that the methodology is presented has been revised. The original supplementary material is now incorporated in the main text. There is no supplementary text in the present draft.

The original §S1 "Coupled Thermal-Kinetic Reaction Calculations" and §S2 "New Parameterization of Thermogenic Kinetic Reaction Modelling Results" are now summarised in the first sub-section of the Results (L111), and the full equations are retained in two corresponding sub-sections of the new Methods (L409, L504).

The original §S3 "Relationship between Generated and Emitted Methane" is now in the Discussion (L376).

The original §S4 "Carbon Emissions from Magma Degassing" and §S5 "Combined Carbon Emissions from the Entire Sill Province" are both summarised in sub-sections of the Results (L162, L207) and the equations given in a sub-sections of the Methods (L578, L659).

The original §S6 "Isotopic Composition of Emissions" is summarised in the Discussion (348) and the equations given in a sub-sections of the Methods (L672).

The original §S7 "Determining Sill Locations" is now a sub-section of the Methods (L738).

*I've a few suggested tweaks/corrections to the supplementary text. (1)
In the third paragraph, correct 'may to overestimate'.*

Done.

(2) I suggest giving the conductive cooling PDE in full, partly because the conductive cooling model is central to the thermal evolution calculations, but also because it contains complicating terms (i.e. the latent heat of crystallization). I think its inclusion would help the reader to understand Figure S4 more easily.

PDE now stated in Methods sub-section "Kinetic reaction modelling of thermogenic emissions". The equation for modification to thermal diffusivity to account for release of latent heat during sill solidification is also now given.

(3) What are the horizontal and vertical lines in Figure S2b showing?

This is Supplementary Fig. 3b of the new draft. Caption now explains these lines.

(4) I think 'blue circles' should be 'blue squares' in the caption to Figure S3b?

This is Fig. 6a of the new draft. Caption corrected to say "squares".

(5) I cannot see the green curves referred to in the caption to Figure S7.

This figure does not appear in the new draft. The carbon isotope information is now included in Fig. 8 (which was Fig. 3 in the previous draft) in line with this reviewer's request for the carbon compositional data to be in the main paper. The green lines appear on Fig. 8a–c.

Reviewer 3: point-by-point response

The paper presents the results of modeling of carbon emissions caused by the North Atlantic Igneous Province causing a significant warming event at the Paleocene-Eocene boundary. The model links the intrusion of sills in Mesozoic sediments by basaltic magma generated by the North Atlantic mantle plume, using information on the speed and relative movement of the plume to constrain the speed of generation of the sill complexes. This information is then used to model the production of carbon gas from the intruded sediments (and the sills?). The carbon excess injected in the atmosphere has been linked to the extreme warming.

The study is interesting as it quantifies and discusses many of the geological variables controlling the final carbon released and the relative timing and tempo of its release. As such it would make a good addition to the growing literature dedicated to this event and also very relevant with respect to the present global warming.

My problem with the paper is that the main body of the paper tends to be very diffuse and not very quantitative. I will discuss some of the examples below. The more substantial part of the paper is the Method section and the Supplementary parts.

The present draft has been substantially reorganised in line with comments from all reviewers and the editor. All of the quantitative information that previously sat in the methods and supplementary sections has been transferred into the main text and methods section of the new manuscript.

The title is much generalized. It indicates the general topic of the paper, but it is not very specific on the substance of the study, specifically the attempt to quantify amount and speed of carbon gas generation to sill complexes and to the mantle plume.

The title has been changed from "Large Igneous Province greenhouse gas flux can initiate Paleocene-Eocene Thermal Maximum" to "Large Igneous Province thermogenic greenhouse gas flux could have initiated Paleocene-Eocene Thermal Maximum climate change". Both the original and new versions contain the word "flux" which clearly indicates that we are quantifying the rate of gas generation over time. The word "flux" also implies a rate of mass or volume change, which is not implied by "speed". The new title contains the word "thermogenic", which is a compact way of saying "contact heating by sills". The most common model for initiation of the Earth's major LIPs is the start-up phase of a mantle plume, and this is the case for the North Atlantic Igneous Province. However, we want to avoid the term "mantle plume" because it implies a deep mantle structure that is not necessary for our theory; we are only concerned with the flux of hot mantle within the asthenosphere, for which there is excellent observational evidence, and do not want to get embroiled in a debate about the depth of origin of this hot material, for which the evidence is less compelling. We use the general term LIP in the title as opposed to NAIP because our methodology can be applied to any other LIP, many of which have both sill provinces and associated climate change events.

The abstract: I had to stop and reflect on the meaning of several sentences. Some examples:

Line 2: 'Initiation of the North Atlantic Igneous Province (NAIP) spanned the PETM...'. Why initiation? It implies that it was followed by a main phase and then by a termination? How is the initiation defined with respect to the period of emplacement of the sills? How long did the PETM last?

The new, shorter abstract does not mention "initiation". The Results sub-section "Derivation of link between τ_{repeat} and mantle plume flux" details the relationship between mantle flow, magma generation and sill intrusion. This section also avoids the term "initiation" and explains that a hot blob of mantle was travelling around a pre-existing convection cell. This section, the new Fig. 1 and the Discussion also clarify that the melting to form the NAIP sill province was associated with arrival of a hot pulse of mantle within a pre-existing convection cell.

Line 5: '... the multi-millennial timeframe of PETM onset'. As above, what is the PETM onset? Was it followed by a main period and then a termination phase?

Line 11: 'the onset period'. Again, what is that?

The new, shortened abstract does not mention PETM onset. Instead it says that the NAIP could have initiated PETM climate change (L28). The sentence in the first paragraph of the Introduction that previously began "During the PETM..." has been altered to begin "During PETM initiation..." (L40) and goes on to define what we and others mean by PETM initiation.

Line 17: 'magma associated carbon source'. Suggests that the carbon is released by the magma, in reality most of the carbon is modelled as coming from the intruded sediment. Nobody would ever guess this by reading the abstract. Thus, I find the abstract mainly very suggestive but very little informative.

This wording was chosen as a compact way of implying that we include both mantle-derived and thermogenic carbon in the modelling. However, the wording of the new shorter abstract avoids this issue, and the word thermogenic is now included in the title because one result of this study is that thermogenic carbon is significantly more important than magma-derived carbon.

Main text, first sentence: 'A temporal association between Large Igneous Provinces (LIPs) and perturbations to global climate and the carbon cycle occurs throughout Mesozoic time...' The statement could be interpreted as this having been a continuous process, with some more specific stronger episodes.

A continuous process with some more specific stronger episodes is exactly what was described in the main text of the previous manuscript in the first paragraph of "Linking Mantle Plume Flux and Sill Intrusion Frequency". In the new manuscript, this description is retained in description sits at L228: "Flow of mantle rock within this and other plumes is unsteady, or "pulsing". The new Fig. 1 also clarifies that the melting to form the NAIP sill province was associated with arrival of a hot pulse of mantle within a pre-existing convection cell.

page 3, line 3: 'a rapid period' time is not rapid or slow. You mean short period?

Agreed; "short" swapped for "rapid" (L42).

Line 4: 'Although the North Atlantic Igneous Province (NAIP) LIP and the PETM are closely coincident in time, the rate of NAIP carbon emissions has not yet been reconciled with the rapid onset of PETM climate change.' Very generalized statement: please state what 'the rate of NAIP carbon emissions' is, and quantify what the 'rapid onset of PETM climate change' actually was (increase of x oC over y years).

Sentence changed to "...the rate and duration of NAIP carbon emissions have not yet been reconciled with the c. 10 kyr onset of PETM climate change" (L43). The change in temperature during PETM onset, the mass of carbon involved and the duration of the onset period have all been quantified in the preceding sentence: "During PETM initiation, release of 0.3–1.1 PgC/yr of carbon as greenhouse gases to the ocean-atmosphere system drove 4–5 C of global warming over a short period (<10,000 years) (L40).

End of second paragraph on page 3: 'However, all these theories struggle because their timeframes have not been shown to match the timeframe of the PETM onset. Consequently, the NAIP has typically been relegated to a driver of longer term background warming, which perhaps triggered more rapid release of carbon from other temperature-sensitive near-surface reservoirs around the globe'. It seems to me that it overgeneralizes. Some of the quoted papers seem to draw quite a direct link between PETM and magmatism. Again a more specific and less generalized presentation should mention the actual arguments and mechanisms proposed in the literature, so the reader would know exactly what the authors refer to.

First sentence mentioned by the reviewer has now been clarified by altering it to "However, all these theories struggle because none has been shown to deliver a peak in gas emissions flux whose duration matches the 10-kyr timeframe of the PETM onset" (L53). The carbon sources and release mechanisms are listed and individually referenced at the start of this paragraph (L46). The intention of the second sentence mentioned by the reviewer is to generalise in order to encompass the various different postulated carbon sources and release mechanisms stated at the start of the paragraph.

3rd line, 3rd para, page 3: '...these sills are the most likely source of the large total mass of carbon...'. The model assumes that the bulk of the C was released from the sediments; just 5% came from the magma. This sentence suggests the opposite.

Agreed. Sentence modified to "We focus on thermogenic methane produced by shallow igneous sills (sub-horizontal sheets of magma) and released to the atmosphere or shallow ocean through hydrothermal vents because this is the most likely source of the large mass of carbon (up to 13,000 PgC) required to explain the entire PETM" (L61).

page 4, line 6: 'cooling time period for the sill, of order 100s to 1000s years...'

Missing reviewer's statement of what is wrong here.

page 4, 2nd para, line 2: Please define 'the PETM onset period'

Now defined in the first paragraph of the introduction (L40).

page 6, 2nd para, line 2: 'The mantle plume flux parameter is a fundamental measure of how fast temperature pulses move round a mantle convection cell'. The variable has a dimension of [surface over time] and seems to me to indicate the progression of the plume with respect to the overlying plate, but a more concrete explanation would certainly be useful if the paper has to be read also by non-specialists.

We deliberately use the general term "mantle plume flux" to introduce the concept (L242) because several different measures of mantle plume flux are used in the geophysics literature, including mass flux, volume flux, area flux and buoyancy flux. They have different dimensions but describe essentially the same processes. We explain the our use of the area flux variant later in the paragraph (L253): "Thus Q can also be interpreted as the mantle plume area flux, equivalent to the rate of change in plume head surface area A_{mantle} ."

These examples reflect the style of the paper. Very often using generalizations that lack more specific references and arguments, and are often very qualitative, lacking quantitative back-up. It is almost like the authors want to keep things as diffuse as possible. Perhaps adding a figure that illustrates the relationships between plume and sills and sediments and rift, pointing out the various parameters used in the model, would help.

We did try to place as many of the details and equations as possible in the supplementary material of the previous draft in order to fit the space requirements of other Nature family journals. In the present manuscript, we have taken advantage of Nat. Comm's more generous formatting requirements and moved more of the quantification and details into the main text. New Fig. 1 now illustrates the relationships between plume and sills and sediments and rift. The revisions to Fig. 4 point out the various parameters used in the model more clearly. New Supplementary Figure 2 shows an example seismic image of sills.

Fig. 2: I suggest enlarging a-c, where now the details and labeling of far too small. I would then remove d-f, the symbols and letter are so small that they are next impossible to decipher. And the movie shows the distribution very well.

I think this comment refers to original Fig. 3, which is Fig. 8 in the new version. We have enlarged a–c and added new panels d–f in response to other reviewer's comments. We have retained enlarged snapshots from the movie as panels g–j because not all first time readers will be in a position to view the supplementary movie.

In chapter 4 of the supplementary material there is a discussion of the C in the magma. It is interesting, but I was also left in some confusion. Since the total C released is important, it is not evident to me why lavas and lower crustal bodies are not included. The reasoning in that paragraph seems to contradict itself a bit. But such information with estimated quantities and length of processes should clearly be given in the main text.

§S4 of the supplementary material of the previous draft has been relocated to the Discussion section of the main text in the new manuscript (L362). Mass information has now been added to the sentence near the end of this paragraph: "The resulting emissions flux of 0.005–0.015 PgC yr⁻¹ and mass of 50–150 PgC from lavas and deep intrusions during the PETM onset period is negligible in comparison with the flux of 0.2–0.5 PgC yr⁻¹ and mass of 2,750 PgC from thermogenic methane and sill-degassed carbon dioxide" (L369).

F. Corfu 2-5-2019

Reviewers' comments:

Reviewer #2 (Remarks to the Author):

Jones et al. have substantially revised their manuscript, and have addressed my comments satisfactorily. I recommend publishing their novel and interesting paper. The restructuring of the paper and the addition and modification of figures has helped to explain their data, approach and results much more clearly. The paper is well written. I have a few minor suggestions/questions they might want to address.

[1] It is not clear to me that forced folds must form at the seabed (e.g. lines 185-186)?

[2] L250-251: Define 'hot', e.g. as excess temperature.

[3] L271: '... along a line connecting...' is an odd way to put it, suggest '... between...'.
[4] L429: typo, '... may to overestimate...'.
[5] L1029: typo, "... from forced".
[6] L1045: typo, '... blue lines...'. Need to define grey and black lines.
[7] L1053: suggest 'triangle⁶; inverted triangle⁵⁴'.
[8] L1058: typo, 'red squares'.
[9] Line 1084: define grey/black curves.
[10] Figure 8. Still very hard to read annotations on pie charts and legend. Please make bigger.
[11] L1150: typo(?), 'param'.

Reviewer #4 (Remarks to the Author):

I did not participate in the initial review of this manuscript, but I agree with the general sense of the first round of review, that this manuscript represents a significant advance in our understanding of the role that thermogenic methane from the NAIP sills played in the greenhouse forcing of the PETM. I also think the authors have done a good job of responding to the previous reviews. So I recommend publication with minor further edits.

I would recommend some expansion of the discussion of the discrepancies between the inverse modeling estimates of emission rates vs. those presented here. Fig. 8a shows a good general agreement, but the initial apparent high peak rates of the inverse model aren't achieved (and that modeling was based on a forcing function from the isotopes that doesn't reach a minimum value until about 25 kyr after onset, an onset that is longer than the <10 ky. the manuscript here describes in the introduction and interpreted by some/many (we, Cui et al., 2011 argued for a longer onset duration, as in the Gutjahr paper, but others disagree; one problem is that the carbonate records all suffer from dissolution during the onset). The authors here gloss over the 0.5 Pg C yr⁻¹ discrepancy between their estimates of peak rates and the model-inversion rates, but that difference is important during the onset (e.g., it might require the positive feedbacks of other carbon releases driven by the thermogenic methane release estimated here). On the other hand, the post-onset decline IS well matched. That also is significant because it suggests that there might not be a need to invoke organic carbon burial to draw down the d13C rapidly enough, as suggested by Cui et al. (2011) and others.

Other minor comments:

Line 53: theories can't struggle, only people struggle

Line 93 through 103: I would recommend using the past tense for activities that clearly occurred before the paper was being written. For example, you clearly aren't assembling the database during the writing or presentation of the paper, nor are you developing a method to determine carbon emissions.

Line 121: Fig. 2 (not Figs 2)

Line 343: I don't understand the point of thermogenic methane carbon being a larger mass proportion of the molecule than CO₂. The estimates are in Pg C, not Pg CH₄ or Pg CO₂. I don't see where the mass of the molecule comes into play.

Lee Kump, Penn State

Point-by-Point Responses to Reviewers

Changes to text highlighted in yellow here and in the revised manuscript file.

Reviewer #2

Jones et al. have substantially revised their manuscript, and have addressed my comments satisfactorily. I recommend publishing their novel and interesting paper. The restructuring of the paper and the addition and modification of figures has helped to explain their data, approach and results much more clearly. The paper is well written. I have a few minor suggestions/questions they might want to address.

[1] It is not clear to me that forced folds must form at the seabed (e.g. lines 185-186)

Sentence now reads, "Depths of intrusion were obtained from the subset of sills for which the coeval seabed can be identified from associated vents and/or onlap of strata onto forced folds²⁶," to clarify that the seabed is recognised by onlap onto the seabed horizon and to provide a reference for this standard interpretation.

[2] L250-251: Define 'hot', e.g. as excess temperature.

Sentence now reads, "If melting occurs across the entire footprint of the expanding patch of unusually hot mantle...", to match the terminology in the preceding paragraphs.

[3] L271: '... along a line connecting...' is an odd way to put it, suggest '... between...'

Sentence now reads, "These studies estimate a parameter k , with units of diffusivity, which characterizes the speed of the locus of peak dynamic support as it travels along a line between the two sedimentary basins."

[4] L429: typo, '... may to overestimate...'

Sentence now reads, "This G value may underestimate the true value in the Vøring and Møre Basins⁵⁰ but may overestimate the true value in the Rockall Basin, where extreme crustal extension has occurred."

[5] L1029: typo, "... from forced'.

Clause now reads, "light grey, sill thickness inferred from forced folds."

[6] L1045: typo, '... blue lines...'. Need to define grey and black lines.

Sentence now reads, "Horizontal blue line and envelope show most likely value^{5,6} and range^{4,5} of peak emissions required to explain PETM onset determined from climate models."

[7] L1053: suggest 'triangle⁶; inverted triangle⁵⁴'.

Clause now reads, "triangles, from forward modelling of carbon isotope composition and deep-sea carbonate dissolution (triange⁶; inverted triangle⁵⁶."

[8] L1058: typo, 'red squares'.

Sentence now reads, "Plume centre locations: red circles (references in Table 1)."

[9] Line 1084: define grey/black curves.

Sentence now reads, "Cumulative proportion of overlapping thermal aureoles (stack of 100 stochastic runs)."

[10] Figure 8. Still very hard to read annotations on pie charts and legend. Please make bigger.

This is a problem with the PNG version of the file embedded in the word document, which have reduced resolution to avoid a very large file size. It should be resolved when larger versions of individual figures are sent in.

[11] L1150: typo(?), 'param'.

Sentence now reads, "Dependence of carbon emissions parameterisation on sill thickness and emplacement depth."

Reviewer #4

I did not participate in the initial review of this manuscript, but I agree with the general sense of the first round of review, that this manuscript represents a significant advance in our understanding of the role that thermogenic methane from the NAIP sills played in the greenhouse forcing of the PETM. I also think the authors have done a good job of responding to the previous reviews. So I recommend publication with minor further edits.

I would recommend some expansion of the discussion of the discrepancies between the inverse modeling estimates of emission rates vs. those presented here. Fig. 8a shows a good general agreement, but the initial apparent high peak rates of the inverse model aren't achieved (and that modeling was based on a forcing function from the isotopes that doesn't reach a minimum value until about 25 kyr after onset, an onset that is longer than the <10 ky. the manuscript here describes in the introduction and interpreted by some/many (we, Cui et al., 2011 argued for a longer onset duration, as in the Gutjahr paper, but others disagree; one problem is that the carbonate records all suffer from dissolution during the onset). The authors here gloss over the 0.5 Pg C yr⁻¹ discrepancy between their estimates of peak rates and the model-inversion rates, but that difference is important during the onset (e.g., it might require the positive feedbacks of other carbon releases driven by the thermogenic methane release estimated here). On the other hand, the post-onset decline IS well matched. That also is significant because it suggests that there might not be a need to invoke organic carbon burial to draw down the d13C rapidly enough, as suggested by Cui et al. (2011) and others.

We have expanded the discussion of the discrepancies between the inverse modelling estimates of emission rates versus our new emissions estimates, and also commented on the timing of peak flux, by adding several sentences to the final paragraph of the

discussion. This paragraph now reads: "We have presented the first mechanistic model of carbon emissions flux from any proposed PETM carbon source that is directly constrained by measurements of the geological structures that control the emissions. Differences between our carbon emissions source function and carbon emissions inferred independently from climatic effects provide for the first time the potential to directly measure climate system feedbacks. However, it is too early to interpret source-sink differences in Fig. 8 in terms of such feedbacks. For example, modelling the carbon sink^{5,6} indicates peak emissions flux of 0.5–0.6 PgC yr⁻¹ followed by a sharp decrease. Our median estimate of the carbon emissions source has a peak 0.2–0.3 PgC yr⁻¹ lower and a smoother post-peak decrease. However, the smoothness of our carbon flux estimate derives partly from the smooth curve used to represent the sill distribution (ρ) measurements (Fig. 7c). Given sparse ρ measurements near the sill province centre, we cannot yet rule out a carbon emissions source function that would match emissions inferred from inverse modelling of the carbon sink more closely (Supplementary Fig. 5). Considering uncertainties in the other controls on τ_{repeat} , peak emissions flux could occur between 1 and 20 kyr (Figs 5 & 8). Thus, more sill distribution measurements, more mantle plume flux measurements and improved measurements of the shape and positioning of the plume head are required to reduce the uncertainty range of our solid Earth emissions model. These combined uncertainties at present leave room for a scenario in which no significant greenhouse gas sources external to the NAIP sill province are required to explain the PETM onset and the first few 10s of thousands of years of the PETM recovery. Clarification of this carbon cycle behaviour will impact modelling and management of future climate change. We therefore eagerly anticipate improvements in modelling both the source and fate of carbon for the NAIP-PETM pair, and also other LIPs paired with major climate change events." To justify these additional sentences, we have added a sentence and equation at L219–222, and a new Methods subsection "Relating sill distribution and carbon emissions" (L748).

In the remainder of the manuscript, we have changed the various references for the duration of PETM initiation from c. 10 kyr to c. 20 kyr, and made reference to Cui et al. (2011), e.g. L40: "During PETM initiation, release of 0.3–1.1 PgC yr⁻¹ of carbon as greenhouse gases to the ocean-atmosphere system^{4–6} drove 4–5°C of global warming⁷ over a short period (<20,000 years)^{5,8–10}" and similar changes in L43, L53, L317.

Other minor comments:

Line 53: theories can't struggle, only people struggle

As a system of ideas, a theory is an expression of people's thought. Thus, the statement "theories struggle" is a compact way of saying that the people who made the theories have struggled to understand the problem, which does conform to the reviewer's rule.

Line 93 through 103: I would recommend using the past tense for activities that clearly occurred before the paper was being written. For example, you clearly aren't assembling the database during the writing or presentation of the paper, nor are you developing a method to determine carbon emissions.

Agreed. Paragraph now reads, "Here we demonstrate for the first time that the NAIP sill province could have intruded sufficiently rapidly to initiate the PETM. We tackle the problem in two stages. First, we determine the τ_{repeat} that would be required for the NAIP sill province to match carbon emissions rates that have been independently shown to initiate the PETM. Secondly, we demonstrate that such τ_{repeat} values were likely achieved during the most intense phase of NAIP sill intrusion. These two steps required development of several new databases and calculation procedures. We began by

developing a new parameterisation of thermogenic and magmatic carbon emissions from individual sill-vent systems of known dimensions intruding a host of known organic content. We then assembled a large new database of NAIP sill and host-rock observations. We also developed a method to determine the carbon emissions from an entire sill province by summing emissions from many sill-vent systems. Together, these components allow Monte Carlo simulations of geologically plausible combined carbon emissions from a sill province when the τ_{repeat} is specified *a priori*, which show that τ_{repeat} of 2–6 yr would be required to initiate the PETM. To complete the second stage in our argument, we developed a novel alternative to dating volcanic products that considers instead the mantle convection process that generated the sill province magma: we derived an expression linking τ_{repeat} to mantle plume flux (Fig. 1). We then assembled new databases of mantle plume flux measurements and the geographical distribution of sills across the NAIP. We use this information to estimate how τ_{repeat} varied throughout the emplacement history of the NAIP sill province, and show that τ_{repeat} could have dropped below 5 yr and initiated the PETM. Thus, we present the first predictive, mechanistic model of carbon emissions flux from a LIP."

Line 121: Fig. 2 (not Figs 2)

Sentence now reads, "We first carried out a series of coupled thermal and reaction kinetic calculations that spanned the observed ranges in sill dimensions and emplacement depth (Fig. 2)."

Line 343: I don't understand the point of thermogenic methane carbon being a larger mass proportion of the molecule than CO₂. The estimates are in Pg C, not Pg CH₄ or Pg CO₂. I don't see where the mass of the molecule comes into play.

Sentence now reads, "First, sill magma typically contains 0.5% CO₂ by mass, or 0.14% carbon, whereas the mean organic carbon content within NAIP sill host rocks frequently exceeds 1% (Fig. 4)." Re-written this way, it duplicates the previous third point, which is now deleted.

Lee Kump, Penn State

REVIEWERS' COMMENTS:

Reviewer #4 (Remarks to the Author):

I feel that the authors have done a good job of addressing my concerns, and recommend publication. Lee Kump, Penn State